# FORWARD LEARNING OF GRAPH NEURAL NETWORKS

**Namyong Park**[1]**, Xing Wang**[1]**, Antoine Simoulin**[1]**, Shuai Yang**[1]**, Grey Yang**[1]**, Ryan Rossi**[2]**,
Puja Trivedi**[3]**, Nesreen Ahmed**[4]
[1]Meta AI, [2]Adobe Research, [3]University of Michigan, [4]Intel Labs

## ABSTRACT

Graph neural networks (GNNs) have achieved remarkable success across a wide
range of applications, such as recommendation, drug discovery, and question an-
swering. Behind the success of GNNs lies the backpropagation (BP) algorithm,
which is the de facto standard for training deep neural networks (NNs). However,
despite its effectiveness, BP imposes several constraints, which are not only bio-
logically implausible, but also limit the scalability, parallelism, and flexibility in
learning NNs. Examples of such constraints include storage of neural activities
computed in the forward pass for use in the subsequent backward pass, and the
dependence of parameter updates on non-local signals. To address these limita-
tions, the forward-forward algorithm (FF) was recently proposed as an alternative
to BP in the image classification domain, which trains NNs by performing two
forward passes over positive and negative data. Inspired by this advance, we pro-
pose FORWARDGNN in this work, a new forward learning procedure for GNNs,
which avoids the constraints imposed by BP via an effective layer-wise local for-
ward training. FORWARDGNN extends the original FF to deal with graph data and
GNNs, and makes it possible to operate without generating negative inputs (hence
no longer forward-forward). Further, FORWARDGNN enables each layer to learn
from both the bottom-up and top-down signals without relying on the backpropa-
gation of errors. Extensive experiments on real-world datasets show the effective-
ness and generality of the proposed forward graph learning framework. We release
our code at `https://github.com/facebookresearch/forwardgnn`.

## 1 INTRODUCTION

Graph neural networks (GNNs) (Zhou et al., 2020; Wu et al., 2021) are a family of deep learning
methods designed to operate on graph-structured data, such as social networks, knowledge graphs,
and citation networks. In recent years, GNNs have achieved state-of-the-art results across a wide
range of applications that involve graphs, such as recommendation (Fan et al., 2019), drug discov-
ery (Zhang et al., 2022), traffic forecasting (Jiang & Luo, 2022), question answering (Park et al.,
2022a; 2020), and program analysis (Allamanis, 2022). Behind the huge success of GNNs, as well
as other types of neural networks, lies the backpropagation (BP) algorithm (Rumelhart et al., 1986;
Plaut et al., 1986), which is the de facto standard for training deep neural networks.

BP adjusts the parameters of a neural network in a way that reduces the discrepancy between the net-
work's output and the target value (*e.g.*, ground-truth class labels). Specifically, these updates by BP
are performed in two passes. First, in the forward pass, the input is fed forward through the network
to produce an output, while storing intermediate activations of the neurons. Then, in the backward
pass, an error between the output and target gets propagated backward through the network using the
chain rule, which uses the stored activations to calculate the gradients of the loss with respect to the
network parameters. Then using gradient descent (Kiefer & Wolfowitz, 1952; Robbins & Monro,
1951), parameters are adjusted in the negative direction of the gradients, thereby reducing the loss.

Spurred by the success of BP, several prior works investigated whether BP could be a model of how
the brain learns (Lillicrap et al., 2020; Scellier & Bengio, 2017; Crick, 1989). However, there is no
evidence that the brain follows the learning constraints imposed by BP, *e.g.*, (i) neural activations
computed in the forward pass need to be stored for use in the backward pass, which increases
memory overhead, (ii) parameter updates depend on the signals from all downstream neurons, while
biological synapses learn locally, *i.e.*, from the signals of locally connected neurons (Whittington
& Bogacz, 2019), and (iii) parameter updates occur only after the forward pass, and in its reverse
order (*i.e.*, from the last down to the first layer). These constraints not only render BP biologically
implausible, but also limit the scalability, parallelism, and flexibility in learning neural networks.

The forward-forward algorithm (FF) (Hinton, 2022) is a representative recent approach designed to be a biologically plausible alternative to BP. FF avoids the above constraints of BP by replacing the forward and backward passes in BP with two forward passes, which operate on the positive and negative data, respectively, with opposite objectives. By adopting a layer-wise forward-only training scheme, FF avoids the restriction of BP that a perfect knowledge of the computations in the forward pass needs to be retained to compute the derivatives for backpropagation. In preliminary experiments on image classification, FF showed comparable classification performance to BP.

The training of GNNs is limited by the same constraints of BP as discussed above, which restrict the scalability and flexibility of model training, as well as rendering the learning process biologically implausible. While FF and its recent variants (Hinton, 2022; Zhao et al., 2023; Lee & Song, 2023; Paliotta et al., 2023) saw promising results, they fail to provide an effective and efficient alternative for learning GNNs for the following reasons. First, the FF algorithm involves generating negative data, which is a task-specific process (*e.g.*, handcrafting masks for images with varying long range correlations, and manipulating images via overlaying class labels), and needs to be defined to be used for new tasks. Existing FF methods are mainly designed for image classification, and thus are difficult or ineffective for handling graph data. Second, the current way FF generates negative data is not scalable as it takes an increasing amount of memory and computation time as more negative samples are used. To date, the potential of FF for the fundamental graph learning (GL) tasks, *i.e.*, node classification and link prediction, has not been studied, and it remains to be explored how effectively FF or alternative forward learning procedures can train GNNs for fundamental GL tasks.

To bridge this gap, we develop FORWARDGNN in this paper, a forward learning framework for GNNs, which builds upon and improves the FF algorithm (Hinton, 2022) in learning GNNs for graph learning tasks. First, FORWARDGNN proposes two orthogonal graph-specific approaches for extending the FF algorithm for GNNs. Importantly, FORWARDGNN removes the need to explicitly construct negative inputs needed for FF to work (*i.e.*, FORWARDGNN requires just a single forward pass, and hence is no longer forward-forward). This greatly improves the efficiency of both training and inference processes. Further, in the local layer-wise training of FORWARDGNN, each layer not only receives input from the lower layer, but also incorporates what the upper layers have learned without relying on the backpropagation of errors, which further improves performance as the bottom-up and top-down signals jointly inform the forward training. We show the effectiveness and generality of the proposed forward graph learning framework in an extensive evaluation using three representative GNNs on five real-world graphs. In summary, we make the following contributions.

- **Forward Graph Learning.** We systematically investigate the potential of forward graph learning, *i.e.*, biologically plausible forward-only learning of graph neural networks for fundamental graph learning tasks, namely, node classification and link prediction.
- **Novel Learning Framework.** We develop FORWARDGNN, a novel forward learning framework for GNNs. FORWARDGNN is agnostic to the message passing schemes of GNNs, and is capable of learning GNNs via a single forward pass, while incorporating top-down signals.
- **Effectiveness.** Extensive experiments show that (1) FORWARDGNN outperforms, or performs on par with BP on link prediction and node classification tasks, while being more scalable in memory usage; and (2) the proposed single-forward approach improves upon the FF-based methods.

## 2 BACKGROUND AND RELATED WORK

### 2.1 GRAPH NEURAL NETWORKS

Given a graph $G = (V, E)$ with nodes $V$ and edges $E$, graph neural networks (GNNs) learn the representation of a node in graph $G$ by repeatedly performing a neighborhood aggregation or message passing over its local neighborhood across multiple layers. Let $\boldsymbol{x}_i \in \mathbb{R}^{F_1}$ be the input features of node $i$, $\boldsymbol{h}_i^{(\ell)} \in \mathbb{R}^H$ be the representations or embeddings of node $i$ learned by layer $\ell$ ($\ell \geq 1$), and $\boldsymbol{e}_{(j,i)} \in \mathbb{R}^{F_2}$ be optional features of the edge $(j, i)$ from node $j$ to node $i$. The message passing procedure that produces the embeddings of node $i$ via the $\ell$-th GNN layer can be described as follows:

$$\boldsymbol{h}_i^{(\ell)} = \gamma^{(\ell)} \left( \boldsymbol{h}_i^{(\ell-1)}, \bigoplus_{j \in \mathcal{N}(i)} \psi^{(\ell)} \left( \boldsymbol{h}_i^{(\ell-1)}, \boldsymbol{h}_j^{(\ell-1)}, \boldsymbol{e}_{(j,i)} \right) \right). \tag{1}$$

Here, $\boldsymbol{h}_i^{(0)}$ is set to $\boldsymbol{x}_i$, $\mathcal{N}(i)$ is the neighbors of node $i$, and $\psi(\cdot)$ is a function that extracts a message for neighborhood aggregation, which summarizes the information of the nodes $i$ and $j$, as well as the

optional edge features $\boldsymbol{e}_{(j,i)}$ if available. $\bigoplus(\cdot)$ denotes a permutation-invariant function (*e.g.*, *mean* or *max*) to aggregate incoming messages, and $\gamma(\cdot)$ is a function that produces updated embeddings of node $i$ by combining node $i$'s embeddings with aggregated messages. With multi-layer GNNs, the embeddings of a node learned via local message passing capture information from its $k$-hop neighbors. This message passing scheme can be considered as generalizing the convolution operator in CNNs to irregularly-structured graph data (Zhou et al., 2020; Daigavane et al., 2021), where connections among nodes are often highly skewed and irregular in contrast to grid-structured images.

Various GNN architectures can be described by this framework, including graph convolutional networks (GCN) (Kipf & Welling, 2017), GraphSAGE (Hamilton et al., 2017), and graph attention networks (GAT) (Velickovic et al., 2018). They differ in terms of how they define the functions in Eq. (1), *i.e.*, $\gamma^{(\ell)}(\cdot)$, $\bigoplus(\cdot)$, and $\psi^{(\ell)}(\cdot)$. For instance, to define the aggregator $\bigoplus(\cdot)$, GraphSAGE examines a few options, such as the mean operator, and a max pooling on top of trainable neighbor transformations, while GAT uses an attention mechanism to perform a learnable weighted averaging. As GNN training is done by BP, it inherits the same constraints of BP discussed in Section 1. In this work, we develop an alternative learning procedure for GNNs without those constraints.

## 2.2 THE FORWARD-FORWARD ALGORITHM

The forward-forward algorithm (FF) (Hinton, 2022) is a greedy layer-wise learning procedure for optimizing multi-layer neural networks (NNs). That is, FF trains layers one at a time, going from the bottom to the top, and trained layers are no longer further optimized when upper layers are trained. The main idea of FF is to replace the forward and backward passes of BP with two forward passes, which operate on the positive (real) and negative (incorrect) data. Given images and their class labels, FF constructs positive samples by overlaying each image with its one-hot encoded labels. Negative samples are constructed in the same way, but using a randomly chosen incorrect label.

**Goodness.** In FF, each layer has its own objective, which is to make the forward pass at the layer assign high goodness to positive data, and low goodness to negative data. As positive and negative data differ only in the included labels, the layer is then trained to ignore the features that do not correlate with the label. Specifically, FF defines the goodness of a layer to be the sum of the squares of the ReLU activations. Let $\boldsymbol{a}_i^{(\ell)}$ denote the ReLU activation that the $\ell$-th layer produces by processing an input vector for object $i$. The goodness $\mathcal{G}_i^{(\ell)}$ of object $i$ by the $\ell$-th layer is computed as

$$\mathcal{G}_i^{(\ell)} = \left\| \boldsymbol{a}_i^{(\ell)} \right\|_2^2. \tag{2}$$

To train each layer to be able to distinguish between positives and negatives using goodness, FF defines the probability that object $i$ is considered to be positive by layer $\ell$ as follows:

$$p(\text{positive}) = \sigma \left( \mathcal{G}_i^{(\ell)} - \theta \right) \tag{3}$$

where $\sigma(a) = 1/(1+e^{-a})$ is the logistic function, and $\theta$ is a threshold, which is to make the goodness be well above the given value for positive data, and well below it for negative data.

**Normalization.** Note that Eq. (3) uses activation's length in L2 norm to compute the goodness. To prevent the next layer from distinguishing positive from negative data by just using the length of the input vector (in L2 norm), FF normalizes the length of the activation before passing it down to the next layer. This way, the next layer is guided to use the orientation of the input vector, not its length.

**Applications and Extensions.** FF was recently used for several applications, *e.g.*, biomedical and hyperspectral image classification (Angulo & Paheding, 2023; Paheding & Angulo, 2023), on-device training (De Vita et al., 2023), imitation learning (Chung et al., 2023), and optical NN training (Oguz et al., 2023). A few studies also presented extensions of FF. Instead of training the layers to predict goodness (Eq. 2), CaFo (Zhao et al., 2023) attaches a class predictor to each layer. In CaFo, the given NNs are randomly initialized and fixed the whole time, and only the layer-wise predictors are locally trained. SymBa (Lee & Song, 2023) adopts an alternative loss to maximize the goodness gap between positive and negative items via direct comparison. The predictive FF (Ororbia & Mali, 2023) combines predictive coding with FF, which jointly learns representation and generative circuits in a biologically plausible manner. In contrast to FF and follow-up works designed for image classification, GFF (Paliotta et al., 2023) adapts FF to graph classification by defining the goodness of a graph. However, GFF is designed specifically for graph classification, and cannot be applied to the two fundamental GL tasks, *i.e.*, node classification and link prediction. Improving upon GFF and earlier works, FORWARDGNN enables forward learning for these GL tasks for the first time.

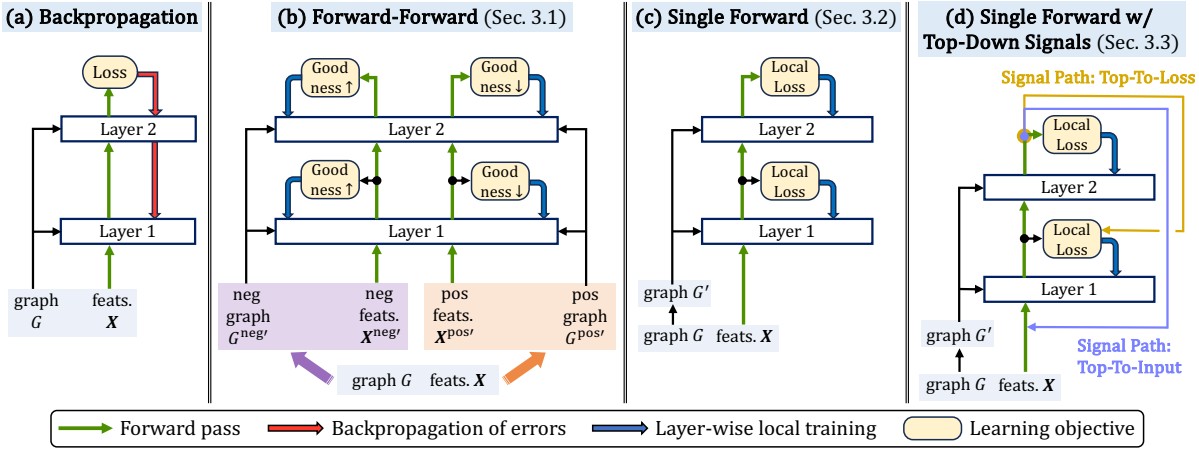

Figure 1: Backpropagation (BP) and proposed forward learning methods for GNNs. (a) BP involves one forward pass, followed by one backward pass through the network. (b) Forward-forward (Sec. 3.1) involves two forward passes on positive and negative inputs. (c) Single forward (Sec. 3.2) learns via just one forward pass. (d) Single forward is extended to incorporate top-down signals (Sec. 3.3).

## 3 FORWARD LEARNING OF GRAPH NEURAL NETWORKS

We develop new forward learning procedures for GNNs in this section. We start by extending the FF algorithm for GNNs, and design a forward graph learning procedure with just a single forward pass. We then extend them so that both the bottom-up and top-down signals can inform the training. Figure 1 provides an overview of the proposed forward graph learning approaches and BP.

### 3.1 ADAPTING THE FORWARD-FORWARD ALGORITHM FOR GRAPH NEURAL NETWORKS

Positive and negative samples play an essential role in FF, as layers are trained to tell them apart in terms of the goodness (Eq. 2). Designed for image classification, FF creates them by overlaying correct or incorrect labels over images. To adapt this idea to deal with graph data and GNNs, we propose two orthogonal approaches for extending node features and graph structure using labels (Fig. 1b).

**Extending Node Features With Node Labels.** In this approach, we transform node labels into auxiliary features and use them to augment node features. Formally, the extended features $\boldsymbol{x}_i'$ of node $i$ is

$$\boldsymbol{x}_i' = [\boldsymbol{x}_i \,\|\, \textit{Label-To-Feat}(i)], \tag{4}$$

where *Label-To-Feat*$(i)$ is a function that maps node $i$'s label to additional features, and $[\boldsymbol{x}\|\boldsymbol{y}]$ denotes a concatenation of vectors $\boldsymbol{x}$ and $\boldsymbol{y}$. Positive and negative features are created by using variants of *Label-To-Feat*$(i)$, which return the one-hot encoding of the correct, and randomly selected incorrect labels for node $i$, respectively. Note that not all nodes have labels, as some nodes are held out for testing, or nodes may have been partially labeled from the beginning. This differs from the image classification setup, where all training images are labeled. For nodes without labels, we define *Label-To-Feat*$(\cdot)$ to be a uniform distribution over the node classes such that all classes contribute equally for these nodes. This approach affects only the node features, while retaining the graph structure. Thus the positive and negative embeddings of training nodes are obtained by running GNNs (Eq. 1) on the same graph $G$ using differently altered features $\boldsymbol{x}_i^{\text{pos}\prime}$ and $\{\boldsymbol{x}_i^{\text{neg},j\prime}\}_j$ as $\boldsymbol{h}_i^{(0)}$.

**Extending Graph Structure With Virtual Nodes.** This approach introduces virtual nodes corresponding to the node classes (*i.e.*, there are as many virtual nodes as the classes), and use the links between real and virtual nodes to make positive and negative samples. For positive samples, each real node is linked to the correct virtual node, that is, the one corresponding to the true label of the node. For negative samples, real nodes are linked with randomly chosen incorrect virtual nodes. Formally, given a set $C = \{c_k\}_{k=1}^K$ of $K$ virtual nodes, the augmented graph $G' = (V', E')$ is defined to be

$$G' = (V \cup C, \;\; E \cup \{(i,c) \mid i \in V_{\text{labeled}}, \; c \in C, \; \textit{Label-RE}(i) = \textit{Label-VR}(c)\}), \tag{5}$$

where $V_{\text{labeled}} \subseteq V$ is the set of labeled nodes, and *Label-RE*$(i)$ and *Label-VR*$(c)$ are functions that map real node $i$ and virtual node $c$ to the class labels, respectively. Specifically, in order to create positive and negative samples, *Label-RE*$(i)$ returns correct, and randomly chosen incorrect labels, respectively, for real node $i$, while *Label-VR*$(c)$ consistently returns the true label for virtual node $c$.

With this augmented graph structure, real nodes can aggregate information from their real neighbors and a virtual node. Similarly, virtual nodes aggregate information from the real nodes they are linked with, which makes them representative of those connected real nodes. *E.g.*, with positive connections, virtual nodes can be considered as class representatives, as virtual nodes are only connected to the real nodes that belong to the virtual node's class. From the perspective of real nodes, the only difference between positive and negative cases is the link to the virtual node. All in all, GNN layers are trained to utilize this difference to assign the goodness (Eq. 2) appropriately, *e.g.*, by ignoring or downplaying some of the aggregated node features that do not correlate well with the node label. The positive and negative embeddings of training nodes are generated by running GNNs (Eq. 1) on differently augmented graphs $G^{\text{pos}\prime}$ and $\{G^{\text{neg},j\prime}\}_j$ with the same input features $\boldsymbol{x}_i = \boldsymbol{h}_i^{(0)}$. Alg. 1 in Appendix C presents the forward learning algorithm discussed in this subsection.

**Inference.** After training GNNs with the above methods, GNN layers can produce the goodness of a node with respect to label $l$, *i.e.*, by running GNNs with node features extended with label $l$ (Eq. 4), or an extended graph where test nodes are linked to the virtual node corresponding to label $l$ (Eq. 5). To predict for a given node, we compute the goodness for all classes using each GNN layer, and accumulate the goodness of all layers. Then the label with the highest accumulated goodness is selected.

## 3.2 FORWARD GRAPH LEARNING VIA A SINGLE FORWARD PASS

Based on FF, the approaches in Section 3.1 operate by learning to distinguish between positive and negative inputs. Specifically, this process involves (1) creating positive and negative inputs in the form of differently augmented graphs or node features, and (2) performing multiple forward passes of GNNs with those augmented graphs and features, which overall has high computational and memory requirements. This limitation is because they incorporate the label information at the *input* level, *i.e.*, by perturbing the inputs to create positive (real) and negative (incorrect) samples, which require separate processing. We tackle this limitation by designing a learning framework that can utilize the label information for the forward GNN training with just a single forward pass (Fig. 1c).

**Framework.** Our main idea is to enable the network to generate learning signals based on the available node labels, instead of requiring positive and negative samples to start the training, which are created by perturbing the input data using labels. In other words, we want the GNN layer to avoid the need for positive and negative samples by generating its own training targets in one forward pass, which can then be used to learn node embeddings that reflect the class structure among the nodes.

To this end, we empower GNN layers to generate class representatives as their learning targets via graph convolution (Eq. 1). Specifically, we turn to our proposed idea of graph augmentation (Eq. 5), and use the positively augmented graph $G^{\text{pos}\prime}$ for the forward pass, in place of the original graph $G$. As discussed in Sec. 3.1, a virtual node in $G^{\text{pos}\prime}$ can be considered to be representative of the nodes in the corresponding class due to its label-based connections to real nodes and the message passing process. This way, GNNs can produce the embeddings of real nodes and class representatives simultaneously in just one forward pass, without having to design a separate architecture to obtain class representatives. As a result, compared to the forward-forward algorithm that involves two forward passes, this framework is no longer forward-forward, and hence a significant departure from FF.

**Learning Signals.** Given the embeddings and labels of real nodes as well as class representatives, we would want the learning signals to train GNNs in such a way that nodes that are similar and in the same class would be close to each other in the embedding space learned by GNNs, while nodes that are dissimilar and in different classes would be away from each other. To train the $\ell$-th GNN layer, we generate such learning signals with the following contrastive learning objective $\mathcal{L}^{(\ell)}$,

$$\mathcal{L}^{(\ell)} = \frac{1}{|V_{\text{labeled}}|} \sum_{i \in V_{\text{labeled}}} \mathcal{L}(\boldsymbol{h}_i^{(\ell)}, i, \ell), \quad \mathcal{L}(\boldsymbol{h}, i, \ell) = -\log \frac{\exp(C(\boldsymbol{h}, \boldsymbol{c}_{Label(i)}^{(\ell)})/\tau)}{\sum_{k \in [\![1,K]\!]} \exp(C(\boldsymbol{h}, \boldsymbol{c}_k^{(\ell)})/\tau)}, \quad (6)$$

where $\boldsymbol{h}_i^{(\ell)}$ and $\boldsymbol{c}_k^{(\ell)}$ are embeddings of node $i$ and class $k$ learned by the $\ell$-th GNN layer; $K$ is the number of node classes; $Label(i)$ maps node $i$ into its class; $C(\cdot, \cdot)$ is a critic function that measures the similarity of two embeddings (*e.g.*, dot product); and $\tau$ is the temperature parameter. This objective produces the learning targets by contrasting real nodes with class representatives, and maximizes the probability of discriminating the correct (node, class) pair among negative unmatched pairs. Alg. 2 in Appendix C presents the algorithm for the proposed single forward approach.

**Inference.** Once GNNs are trained, GNN layers can produce the probability of a node belonging to each class via the softmax inside the logarithm of Eq. (6). To predict for the given node, we compute the class distribution across all GNN layers, and select the label with the highest average probability.

## 3.3 Incorporating Top-Down Signals

In the above forward learning frameworks (Secs. 3.1 and 3.2), GNN layers are trained progressively from the bottom to the top, *i.e.*, once a layer is trained, it remains fixed while upper layers are trained. As a result, when layer $\ell \in [\![1, L]\!]$ is trained, it can only use the bottom-up signals from lower layers 1 to $\ell - 1$, but not the top-down signals from upper layers $\ell + 1$ to $L$. By contrast, with BP, layers can also learn from the top-down signals via the backpropagation of errors throughout the network. Inspired by recent works on the part-whole hierarchy representation in NNs (Hinton, 2021; 2022) and collaborative FF (Lorberbom et al., 2023), we extend the proposed single forward pass approach (Sec. 3.2) to incorporate the top-down signals via two paths (Fig. 1d).

**Signal Path: Top-To-Input.** This path allows to incorporate the output of upper layers into the input of a GNN layer. We now consider the forward pass training to be done over time, such that the training of GNN layers at time $t$ incorporates the top-down signals from the previous time step $t - 1$. Let $\boldsymbol{h}_{i,t}^{(\ell-1)}$ be the embeddings of node $i$ produced by layer $\ell - 1$ at time $t$. The output of upper layers at the previous time $t - 1$, *i.e.*, $\{\boldsymbol{h}_{i,t-1}^{(\ell+1)}, \ldots, \boldsymbol{h}_{i,t-1}^{(L)}\}$, constitutes top-down signals for node $i$. Then

$$\boldsymbol{h}_{i,t}^{(\ell-1)\prime} = Merge(\boldsymbol{h}_{i,t}^{(\ell-1)}, \{\boldsymbol{h}_{i,t-1}^{(\ell+1)}, \ldots, \boldsymbol{h}_{i,t-1}^{(L)}\}) \tag{7}$$

is given to layer $\ell$ as input instead of $\boldsymbol{h}_{i,t}^{(\ell-1)}$. Several options exist for $Merge(\cdot)$, *e.g.*, all upper layers, or an immediate upper layer alone may provide top-down signals, which are averaged and concatenated with $\boldsymbol{h}_{i,t}^{(\ell-1)}$. This way, each layer can learn from the representations learned by upper layers.

**Signal Path: Top-To-Loss.** This path allows to incorporate the output from upper layers into the output of a GNN layer to be used in the loss computation and the inference step. Specifically, the following loss replaces Eq. (6) so that both bottom-up and top-down signals can inform the training,

$$\mathcal{L}_t^{(\ell)\prime} = \Big( \sum_{i \in V_{\text{labeled}}} \mathcal{L}(\boldsymbol{h}_{i,t}^{(\ell)\prime\prime}, i, \ell) \Big) \Big/ |V_{\text{labeled}}| \quad \text{where} \quad \boldsymbol{h}_{i,t}^{(\ell)\prime\prime} = Merge(\boldsymbol{h}_{i,t}^{(\ell)}, \{\boldsymbol{h}_{i,t-1}^{(n)}\}_{n=\ell+1}^{L}) \tag{8}$$

which uses augmented embeddings $\boldsymbol{h}_{i,t}^{(\ell-1)\prime\prime}$ that incorporate top-down signals, instead of $\boldsymbol{h}_{i,t}^{(\ell-1)}$. Via this signal path, each layer can learn to generate outputs in light of what the upper layers learned.

Importantly, even with these signal paths, no gradient flows through GNN layers. Thus, the learning procedure remains to be forward only. Alg. 3 in App. C lists steps for top-down signal incorporation.

## 3.4 Application to Link Prediction

The ideas proposed in Secs. 3.1 and 3.2 use the label information (*e.g.*, label-based virtual nodes), with a focus on the node classification task. Here we discuss that the proposed framework can be naturally used for the forward learning of the link prediction (LP) task. We note that the standard procedure for training and evaluating GNNs for LP shares a lot of similarities with the FF algorithm, and is thus readily adaptable for use in the forward learning framework. Given embeddings $\boldsymbol{h}_i$ and $\boldsymbol{h}_j$ of nodes $i$ and $j$, the link probability $p_{i,j}$ between nodes $i$ and $j$ is modeled as follows:

$$p_{i,j} = LinkProb(EdgeEmb(\boldsymbol{h}_i, \boldsymbol{h}_j)), \tag{9}$$

where $EdgeEmb(\boldsymbol{h}_i, \boldsymbol{h}_j)$ is a function that returns the embedding of edge $(i, j)$ (*e.g.*, element-wise product), and $LinkProb(\cdot)$ transforms the edge embedding into a probability (*e.g.*, summation followed by a sigmoid function). The training goal is to learn to distinguish between the existing (positive) and nonexistent (negative) edges in terms of the estimated link probability. This process is similar to FF in the sense that both aim to tell positive and negative samples apart using the learned scores (*i.e.*, link probability and goodness). Differently from the FF applied for node classification (Fig. 1b), positive and negative edge embeddings can be generated via a single forward pass using the standard training process for LP. Notably, this forward learning approach differs from the BP-based LP training in that each layer is locally trained without backpropagation of errors. The idea in Sec. 3.3 can also be used to incorporate top-down signals in learning the embeddings of nodes, and hence the embeddings of edges. The forward learning algorithm for LP is given in Alg. 4 in App. C.

**Inference.** Once GNNs are trained, each GNN layer produces the probability of a given edge. We compute the link probability to be the average of the link probabilities estimated by all layers.

## 4 EXPERIMENTS

### 4.1 EVALUATION SETUP

Our goal is to systematically evaluate the effectiveness and generality of the proposed forward graph learning approach on fundamental graph learning tasks, namely, node classification and link prediction, using representative GNN models and real-world graphs.

**Node Classification.** Given a graph where each node is assigned to a single class, the task is to predict the class of the given node. We randomly generate the train-validation-test node splits with a ratio of 64%-16%-20%, and evaluate the performance in terms of classification accuracy.

**Link Prediction.** The task is to predict whether the given edge is positive (existing) or not. We split the edges randomly into train-validation-test sets, with a ratio of 64%-16%-20%, which form positive edge sets. We randomly select the same amount of nonexistent edges as the positive edges, which form negative edge sets. We obtain the link probability (Eq. 9) via dot product of the two nodes' embeddings, followed by sigmoid. We measure the performance using the ROC-AUC score.

For both tasks, we perform evaluation using five randomly generated splits, and measure the average and standard deviation of the corresponding performance metric.

**Datasets.** We use five real-world graphs drawn from three domains: PUBMED, CITESEER, and CORAML are citation networks; AMAZON is a co-purchase network; GITHUB is a followership graph. Table 2 in Appendix A provides the summary statistics of these graphs. Appendix A also presents a more detailed description of these datasets.

**Graph Neural Networks.** We evaluate FORWARDGNN and BP using three representative GNNs, namely, graph convolutional network (GCN) (Kipf & Welling, 2017), GraphSAGE (SAGE) (Hamilton et al., 2017), and graph attention network (GAT) (Velickovic et al., 2018).

Experimental settings, *e.g.*, details of GNNs and software used in this work, are given in Appendix B.

### 4.2 COMPARISON WITH THE BACKPROPAGATION ALGORITHM

We compare the proposed forward learning method with backpropagation (BP) using two criteria, namely, (1) task-specific performance (*e.g.*, accuracy) and (2) GPU memory usage increase as more layers are used. Among several forward graph learning approaches we develop in Sec. 3, we report the results obtained with the single-forward learning approach (SF) (Sec. 3.2) in this subsection. Also, for link prediction, we apply the cross entropy loss to the SF method. We later present a comparison among different forward learning approaches in Sec. 4.3.

**Result Visualization.** In Figure 2, we show the task-specific performance (y-axis) and the memory usage of multi-layer GNN compared to the 1-layer GNN (x-axis), obtained with training GNNs with the proposed SF and BP. In Figure 2, the shape and color of a symbol indicate the type of GNNs and dataset, respectively, and symbols that are filled and empty, respectively, denote that the corresponding GNN was trained with BP and SF. For ease of comparison, the filled and empty symbols corresponding to the same GNN on the same dataset are linked by a dotted line.

**Node Classification** (Figure 2a). Using this visualization, we make the following observations.

- ***The proposed forward learning method (SF) performs similarly to BP in many cases*** (which correspond to the near horizontal lines), ***or even outperforms BP in several cases*** (corresponding to the lines that go towards the lower right corner), across different GNNs and graphs.
- ***As we use GNNs with a larger number of layers, BP requires more memory for training*** (the horizontal distance between the filled and empty symbols gets longer), ***while the memory required by SF remains nearly the same*** (empty symbols are located near the left end of the figures). The memory usage increase when using BP goes up to $18\times$ as we use four layers.

**Link Prediction** (Figure 2b). With the same visualization, we have the following observations.

- ***The proposed SF mostly achieves a higher link prediction performance than BP*** (most lines go towards the lower right corner). This is mainly because the link prediction performance of the GNNs trained with BP declines as more GNN layers are used, while the performance of GNNs trained with SF improves to varying degrees (or remains nearly the same) as more layers are used.
- As in the node classification result, ***BP incurs more memory usage as the number of layers increases, while the amount of memory required by SF remains the same***.

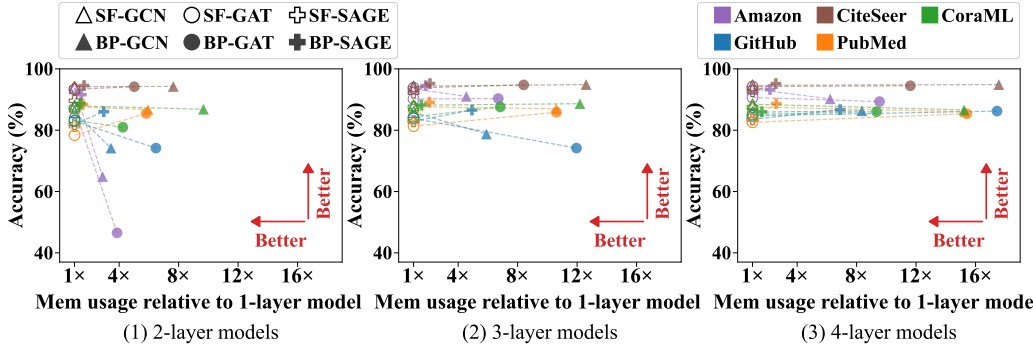

(a) **Node classification** accuracy (y-axis) vs. memory usage increase (x-axis). The proposed single-forward graph learning (SF, empty symbols) performs similarly to backpropagation (BP, filled symbols) in many cases (*i.e.*, near horizontal lines), or even better than BP in several cases (*i.e.*, lines going towards the lower right corner), with no increase in memory usage as more layers are used. Figure 5 shows these results per GNN.

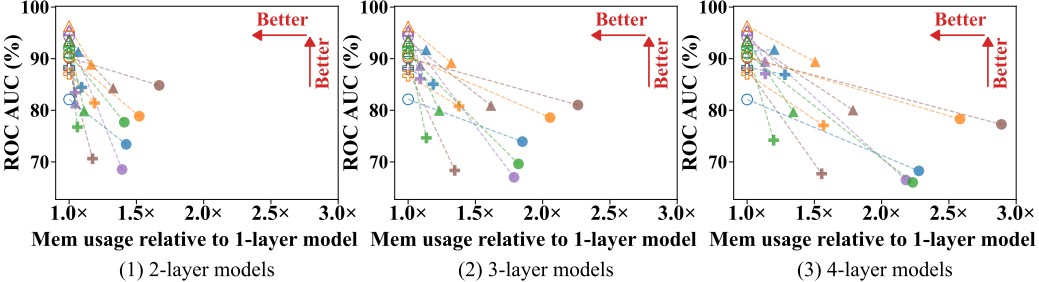

(b) **Link prediction** performance (y-axis) vs. memory usage increase (x-axis). The proposed SF outperforms backpropagation (BP, filled symbols) (corresponding to most lines going towards the lower right corner), with no increase in memory usage as more layers are used. Figure 6 shows these results per GNN.

Figure 2: The single-forward approach (SF, empty symbols) vs. backpropagation (BP, filled symbols) on (a) node classification and (b) link prediction. Lines towards the lower right corner (SF outperforms BP), and upper right corner (BP outperforms SF); horizontal lines (both perform the same).

**Additional Results.** Figures 5 and 6 show the results of Figures 2a and 2b in detail, per GNN model.

### 4.3 COMPARISON AMONG FORWARD LEARNING APPROACHES

In this section, we evaluate the proposed forward learning approaches (Sec. 3) and existing forward learning methods, in terms of the task performance and memory usage.

**Node Classification.** We evaluate the following forward learning methods: (1) **FF-VN** and (2) **FF-LA**, the forward-forward (FF) graph learning approaches (Sec. 3.1), which augment graph data with virtual nodes, and label-appending technique, respectively; (3) **SF**, the single-forward graph learning method (Sec. 3.2); (4) **SF-TopDown**, the SF method with the top-to-input signal path (Sec. 3.3); (5) **FF-SymBa** (Lee & Song, 2023) and (6) **CaFo** (Zhao et al., 2023), which are two extensions of the FF algorithm (Hinton, 2022); and (7) **PEPITA** (Dellaferrera & Kreiman, 2022), a prior state-of-the-art forward learning method preceding FF. Note that FF-SymBa is not directly applicable to node classification, so we adapted it to operate on top of the proposed virtual node method (Sec. 3.1).

Figure 3 shows classification accuracy (y-axis) versus memory usage (x-axis), as GNNs (denoted by the symbol shape) are trained with the above learning methods (denoted by the symbol color). In this section, we report results obtained with two-layer GNN models (Figures 3 and 4 and Table 1).

Table 1: Top-down signals improve classification results. Average classification accuracy obtained with SF and SF-TopDown.

| Method | CITESEER | CORAML | PUBMED | AMAZON | GITHUB |
|---|---|---|---|---|---|
| SF | 91.14 | 84.95 | 81.90 | 89.69 | 83.16 |
| SF-TopDown | **93.58** | **86.01** | **82.21** | **92.09** | **83.68** |

- *The best classification accuracy is achieved by either the proposed single-forward approaches* (SF and SF-TopDown) *or forward-forward (FF) methods* (FF-VN, FF-LA, and FF-SymBa).
- However, ***FF methods (FF-VN, FF-LA, and FF-SymBa) are often unstable***, and are significantly outperformed by the single-forward (SF) approaches. Also, ***they require much larger memory than the SF methods***, as they need to construct multiple negative inputs to train GNNs.

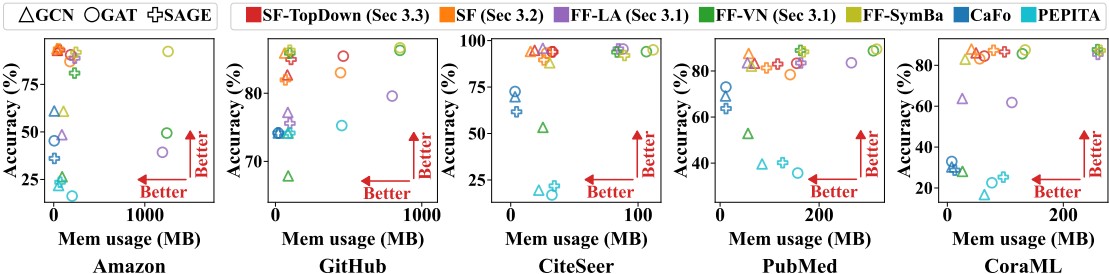

Figure 3: **Node classification** accuracy (y-axis) and memory usage (x-axis) as three GNNs (denoted by symbol shape) are trained with different forward learning approaches (denoted by symbol color).

- CaFo uses the least amount of memory, but performs significantly worse than the proposed methods. This is because CaFo optimizes only the layer-wise predictors, but not the GNN model itself. Also, PEPITA fails to train GNNs effectively for node classification.
- *Incorporating top-down signals enables SF to perform node classification more accurately* as shown in Table 1, which reports the classification accuracy on five graphs, averaged over different GNN models and varying number of layers.

**Link Prediction.** We evaluate the following forward learning methods, which are applicable for link prediction: (1) **ForwardGNN-CE**, (2) **ForwardGNN-FF**, and (3) **ForwardGNN-SymBa**, which are all based on the proposed single-forward framework (Sec. 3.4), while using different learning objectives (*i.e.*, binary cross entropy (BCE), forward-forward objective (Eq. 3), and the objective in (Lee & Song, 2023), respectively); and (4) **CaFo** (Zhao et al., 2023), which uses layer-wise link predictors trained with BCE. Other forward learning methods used above (*e.g.*, PEPITA) are not applicable for link prediction as they are designed specifically for node classification.

Fig. 4 shows the link prediction result (y-axis) and memory usage (x-axis) of different GNNs.

- *The proposed forward learning methods*, especially ForwardGNN-CE and ForwardGNN-SymBa, *consistently outperform CaFo by a large margin*. Freezing the GNN layers is too stringent a restriction for CaFo to effectively learn to do link prediction.
- *ForwardGNN-CE shows the overall best results considering both the accuracy and stability of predictions*. ForwardGNN-SymBa often suffers from high variability in the prediction accuracy.

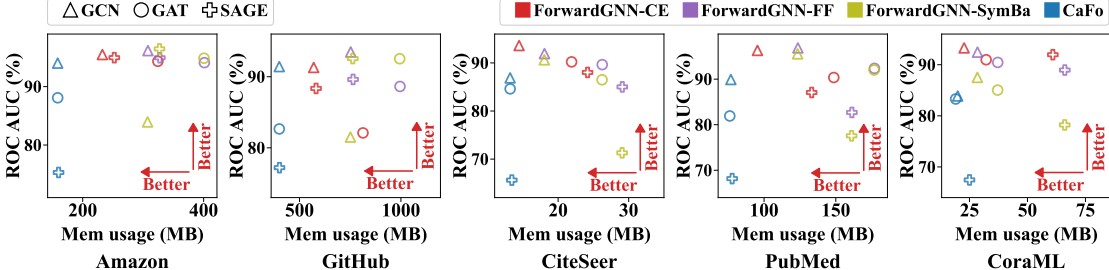

Figure 4: **Link prediction** performance (y-axis) and memory usage (x-axis) as three GNNs (denoted by symbol shape) are trained with different forward learning approaches (denoted by symbol color).

**Further Results.** We present the performance and memory usage for all learning methods, GNNs, and graphs in Tables 3 and 4 for node classification, and in Tables 5 and 6 for link prediction.

## 5 CONCLUSION

Backpropagation (BP) has been the standard algorithm for training graph neural networks (GNNs). However, despite its effectiveness, BP introduces several constraints in learning GNNs. In this work, we investigate the potential of forward-only training in the context of graph learning, and propose FORWARDGNN, a forward learning framework for GNNs. Overall, our contributions are as follows.

- **Forward Graph Learning.** We systematically investigate the potential of biologically plausible forward learning of GNNs for fundamental GL tasks, *i.e.*, node classification and link prediction.
- **Novel Learning Framework.** We develop FORWARDGNN, a novel forward learning framework for GNNs. FORWARDGNN can be used with GNNs with different message passing schemes.
- **Effectiveness.** Extensive experiments show that (1) FORWARDGNN outperforms, or performs on par with BP on link prediction and node classification tasks, while being more scalable in memory usage; and (2) the proposed single-forward approach improves upon the FF-based methods.

## ETHICS STATEMENT

This paper develops a new learning framework for graph neural networks. To evaluate the proposed framework, we use public datasets, which do not contain any personally identifiable information or offensive materials to the best of our knowledge. Also, experiments in this work focus on evaluating the prediction accuracy and memory usage, and do not involve human subjects. We do not foresee any ethical concerns with this work.

## REPRODUCIBILITY STATEMENT

For reproducibility, we describe our experimental settings in Section 4.1 and Appendices A and B, including details of data splitting, evaluation tasks and metrics (Sec. 4.1); dataset description and statistics (App. A and Table 2); and hyperparameter settings, software, and hardware specifications (App. B). All datasets used in this work are publicly accessible (Sec. 4.1). We release our code at `https://github.com/facebookresearch/forwardgnn`.

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

In the appendix, we provide the dataset description (Appendix A), experimental settings (Appendix B), algorithms of the proposed forward graph learning framework FORWARDGNN (Appendix C), discussion on the extensibility of FORWARDGNN to other graph learning tasks (Appendix D), additional related work (Appendix E), and additional experimental results (Appendix F).

## A  DATASET DESCRIPTION

In experiments, we use the following five real-world graphs drawn from three different domains. PUBMED, CITESEER, and CORAML are citation networks from Bojchevski & Günnemann (2018), where nodes represent documents, and edges denote citations between the corresponding papers. Node labels denote the topic of the paper, and node features correspond to the bag-of-words representa-

Table 2: Summary of the graph datasets.

| Dataset | # Nodes | # Edges | # Features | # Classes |
|---|---|---|---|---|
| PUBMED | 19,717 | 88,648 | 500 | 3 |
| CITESEER | 4,230 | 10,674 | 602 | 6 |
| CORAML | 2,995 | 16,316 | 2,879 | 7 |
| AMAZON | 7,650 | 238,162 | 745 | 8 |
| GITHUB | 37,700 | 578,006 | 128 | 2 |

tion of the paper. AMAZON is a co-purchase network from Shchur et al. (2018), which corresponds to the Photo segment, where nodes represent goods, and edges denote that two goods are frequently bought together. Node labels represent item categories, and node features denote the bag-of-words representation of the product review. GITHUB is a followership graph from Rozemberczki et al. (2021), where nodes are developers, and edges denote mutual follower relationships. Nodes are classified into two groups of web and ML developers, and node features denote the developer's information, such as employer name and email address. Table 2 provides the statistics of these graphs.

## B  EXPERIMENTAL SETTINGS

**Hyperparameter Settings, Model Configurations, and Evaluation Details.** Following the default configurations, we ran graph convolutional network (GCN) (Kipf & Welling, 2017) with a normalized adjacency matrix, *i.e.*, self-loops were added and symmetric normalization was applied. We used the mean aggregator with GraphSAGE (SAGE) (Hamilton et al., 2017). For graph attention network (GAT) (Velickovic et al., 2018), we used four attention heads, and concatenated the multi-head attention. Also, based on the default settings of GAT, we added self-loops to the graph, and used a negative slope of 0.2 for LeakyReLU nonlinearity. For all GNNs and across all graph learning tasks, we set the size of hidden units to 128. We used the Adam optimizer with a learning rate of 0.001, and a weight decay of 0.0005. For both tasks, we split the data (*i.e.*, nodes and edges) randomly into train-validation-test sets, with a ratio of 64%-16%-20%. Also, we set the max training epochs to 1000, and applied validation-based early stopping with a patience of 100, for both tasks. In the node classification tasks, some learning algorithms, such as the Forward-Forward algorithm (FF), explicitly construct negative samples for training. For those algorithms, we used all available classes to create negative samples for each node, *i.e.*, given $K$ node classes, $K-1$ negative samples were used for the training of each node. For memory usage, we measured the memory used by the GPU for loading the model parameters, activations, gradients, and the first and second moments of the gradients. For FF and its variants, we set the threshold parameter $\theta$ to 2.0, and for FF-SymBa, we set its $\alpha$ parameter to 4.0, following the settings in the original papers. We set the temperature parameter $\tau$ to 1.0 in Eq. (6). We define the *Merge* function in Eq. (7) to incorporate the signal from the immediate upper layer, *i.e.*, $h_{i,t-1}^{(\ell+1)}$, and concatenate it with the input $h_{i,t}^{(\ell-1)}$. For Eq. (8), we define the *Merge* function to incorporate signals from upper layers, which affect only the loss, but are not fed into the input for the next layer. To apply FF-SymBa (Lee & Song, 2023) for node classification, we used FF-SymBa on top of the virtual node technique (Sec. 3.1) so it can be used for GNNs.

**Software.** We implemented FORWARDGNN in Python 3.8, using PyTorch (Paszke et al., 2019) v1.13.1. We build upon the implementation of the forward-forward algorithm for image classification provided by Nebuly[1]. We used PyTorch Geometric (Fey & Lenssen, 2019) v.2.2.0 for the implementations of GCN, SAGE, and GAT, and for the module that performs edge splitting and sampling used in link prediction. For PEPITA (Dellaferrera & Kreiman, 2022), we adapted the original implementation[2] for GNNs, *i.e.*, after performing forward passes through GNN layers, we compute the error with respect to node labels, and use it to construct modulated inputs for the training nodes.

**Hardware.** Experiments were performed on a Linux server running CentOS 9, with an NVIDIA H100 GPU, AMD EPYC 9654 96-Core Processors, and 2.2TB RAM.

---

[1] https://github.com/nebuly-ai/nebuly
[2] https://github.com/GiorgiaD/PEPITA

## C   ALGORITHMS

We present the algorithms of the proposed forward graph learning approach for node classification, *i.e.*, the forward-forward learning approach (Alg. 1), the single-forward learning approach (Alg. 2), and the single-forward learning approach with top-down signal incorporation (Alg. 3). We then present the forward learning algorithm for link prediction (Alg. 4).

---

**Algorithm 1:** Forward-Forward Learning of GNNs for Node Classification (Section 3.1)

**Input:** GNN $\mathcal{G}$, graph $G$, input features $\boldsymbol{X}$, node labels $\boldsymbol{y}$
**Output:** trained GNN $\mathcal{G}$

/* Construct positive and negative inputs                                                      */

1 **if** *extending graph $\mathcal{G}$ with virtual nodes* **then**
2     $G^{\text{pos}\prime}, G^{\text{neg}\prime} \leftarrow$ positive and negative graphs built using virtual nodes and labels $\boldsymbol{y}$ (Sec. 3.1)
3     $\boldsymbol{X}^{\text{pos}\prime}, \boldsymbol{X}^{\text{neg}\prime} \leftarrow \boldsymbol{X}, \boldsymbol{X}$
4 **else if** *extending input features $\boldsymbol{X}$ by appending labels* **then**
5     $G^{\text{pos}\prime}, G^{\text{neg}\prime} \leftarrow G, G$
6     $\boldsymbol{X}^{\text{pos}\prime}, \boldsymbol{X}^{\text{neg}\prime} \leftarrow$ positive and negative features built by appending labels using $\boldsymbol{y}$ (Sec. 3.1)
7 **end**

/* Train layer by layer progressively from the bottom to the top                               */

8 $\boldsymbol{H}^{\text{pos}\prime}, \boldsymbol{H}^{\text{neg}\prime} \leftarrow \boldsymbol{X}^{\text{pos}\prime}, \boldsymbol{X}^{\text{neg}\prime}$
9 **for** *layer* **in** GNN $\mathcal{G}$ 's layers **do**
10     **while** *not converged* **do**
11        $\boldsymbol{H}^{\text{pos}\prime}_{\text{fwd}} \leftarrow$ *layer*.forward($G^{\text{pos}\prime}, \boldsymbol{H}^{\text{pos}\prime}$)
12        $\boldsymbol{H}^{\text{neg}\prime}_{\text{fwd}} \leftarrow$ *layer*.forward($G^{\text{neg}\prime}, \boldsymbol{H}^{\text{neg}\prime}$)
13        Compute the local loss (*e.g.*, the FF loss (Eq. 3)) for *layer* using $\boldsymbol{H}^{\text{pos}\prime}_{\text{fwd}}$ and $\boldsymbol{H}^{\text{neg}\prime}_{\text{fwd}}$
14        Update *layer*'s parameters using an optimizer of choice, *e.g.*, SGD and Adam
15     **end**
16     *best_layer* $\leftarrow$ load the layer with the best params. found during the local training (lines 10 to 15)
17     **if** *extending graph $\mathcal{G}$ with virtual nodes* **then**
18        $\boldsymbol{H}^{\text{pos}\prime}, \boldsymbol{H}^{\text{neg}\prime} \leftarrow$ *best_layer*.forward($G^{\text{pos}\prime}, \boldsymbol{H}^{\text{pos}\prime}$), *best_layer*.forward($G^{\text{pos}\prime}, \boldsymbol{H}^{\text{pos}\prime}$)
19     **else if** *extending input features $\boldsymbol{X}$ by appending labels* **then**
20        $\boldsymbol{H}^{\text{pos}\prime}, \boldsymbol{H}^{\text{neg}\prime} \leftarrow$ *best_layer*.forward($G^{\text{pos}\prime}, \boldsymbol{H}^{\text{pos}\prime}$), *best_layer*.forward($G^{\text{neg}\prime}, \boldsymbol{H}^{\text{neg}\prime}$)
21     **end**
22     $\boldsymbol{H}^{\text{pos}\prime}, \boldsymbol{H}^{\text{neg}\prime} \leftarrow \boldsymbol{H}^{\text{pos}\prime}$.detach(), $\boldsymbol{H}^{\text{neg}\prime}$.detach()
23 **end**

---

**Algorithm 2:** Single-Forward Learning of GNNs for Node Classification (Section 3.2)

**Input:** GNN $\mathcal{G}$, graph $G$, input features $\boldsymbol{X}$, node labels $\boldsymbol{y}$
**Output:** trained GNN $\mathcal{G}$

1 $G' \leftarrow$ a positively augmented graph using virtual nodes and node labels $\boldsymbol{y}$ (Section 3.2)

/* Train layer by layer progressively from the bottom to the top                               */

2 $\boldsymbol{H} \leftarrow \boldsymbol{X}$
3 **for** *layer* **in** GNN $\mathcal{G}$ 's layers **do**
4     **while** *not converged* **do**
5        $\boldsymbol{H}_{\text{fwd}} \leftarrow$ *layer*.forward($G', \boldsymbol{H}$)
6        Compute the local loss given by Eq. (6) for *layer* using $\boldsymbol{H}_{\text{fwd}}$
7        Update *layer*'s parameters using an optimizer of choice, *e.g.*, SGD and Adam
8     **end**
9     *best_layer* $\leftarrow$ load the layer with the best parameters found during the local training (lines 4 to 8)
10     $\boldsymbol{H} \leftarrow$ *best_layer*.forward($G', \boldsymbol{H}$)
11     $\boldsymbol{H} \leftarrow \boldsymbol{H}$.detach()
12 **end**

---

Note that in Alg. 3 with the top-to-input signal path, we set the top-down signal for the topmost hidden layer to come from the context vector (we used one hot encoding corresponding to the node

label for training nodes, and a uniform distribution vector of the same size for other nodes), following Hinton (2022). Also, in updating the GNN layers with the incorporation of top-down signals, we support both synchronous and asynchronous updates. With asynchronous (*i.e.*, alternating) updates, even-numbered layers are updated based on the activities of odd-numbered layers, and then odd-numbered layers are updated based on the new activities of even-numbered layers. With synchronous updates, updates to all layers occur simultaneously.

---

**Algorithm 3:** Single-Forward Learning of GNNs with Top-Down Signal Paths for Node Classification (Section 3.3)

---

**Input:** GNN $\mathcal{G}$, graph $G$, input features $\boldsymbol{X}$, node labels $\boldsymbol{y}$
**Output:** trained GNN $\mathcal{G}$

1   $G' \leftarrow$ a positively augmented graph using virtual nodes and node labels $\boldsymbol{y}$ (Section 3.2)

2   Initialize $\{\boldsymbol{H}_{t-1}^{(\ell)}\}_\ell$ and $\{\boldsymbol{H}_t^{(\ell)}\}_\ell$ // to store embeddings at the previous and current time steps

     /* Forward-only training with top-down signal incorporation                   */

3   **while** *not converged* **do**

4      **for** $\ell$, *layer* **in** enumerate(GNN $\mathcal{G}$ 's layers) **do**

         // layer index $\ell$ starts from 1

5         **if** *signal-path = "top-to-input"* **then**

6            Construct $\boldsymbol{H}_t^{(\ell-1)\prime}$ by applying Eq. (7) with $\{\boldsymbol{H}_{t-1}^{(\ell)}\}_\ell$ and $\{\boldsymbol{H}_t^{(\ell)}\}_\ell$

7         **else**

8            $\boldsymbol{H}_t^{(\ell-1)\prime} \leftarrow \boldsymbol{H}_t^{(\ell-1)}$

9         **end**

10       $\boldsymbol{H}_t^{(\ell)} \leftarrow$ *layer*.forward($G'$, $\boldsymbol{H}_t^{(\ell-1)\prime}$).detach()

11         **if** *signal-path = "top-to-loss"* **then**

12            Compute the local loss given by Eq. (8) for *layer* using $\{\boldsymbol{H}_{t-1}^{(\ell)}\}_\ell$ and $\{\boldsymbol{H}_t^{(\ell)}\}_\ell$

13         **else**

14            Compute the local loss given by Eq. (6) for *layer* using $\boldsymbol{H}_t^{(\ell)}$

15         **end**

16       Update *layer*'s parameters using an optimizer of choice, *e.g.*, SGD and Adam

17      **end**

18      **foreach** $\ell \in [\![1, L]\!]$ **do**

19       $\boldsymbol{H}_{t-1}^{(\ell)} \leftarrow \boldsymbol{H}_t^{(\ell)}$

20      **end**

21   **end**

---

**Algorithm 4:** Single-Forward Learning of GNNs for Link Prediction (Section 3.4)

---

**Input:** GNN $\mathcal{G}$, graph $G$, input features $\boldsymbol{X}$
**Output:** trained GNN $\mathcal{G}$

1   $E^{\text{pos}\prime}, E^{\text{neg}\prime} \leftarrow$ positive and negative edges sampled from graph $G$ (Section 3.4)

     /* Train layer by layer progressively from the bottom to the top                  */

2   $\boldsymbol{H} \leftarrow \boldsymbol{X}$

3   **for** *layer* **in** GNN $\mathcal{G}$ 's layers **do**

4      **while** *not converged* **do**

5       $\boldsymbol{H}_{\text{fwd}} \leftarrow$ *layer*.forward($G$, $\boldsymbol{H}$)

6       Compute the local loss for *layer* using the link probability (Eq. 9) with node embeddings $\boldsymbol{H}_{\text{fwd}}$, and positive and negative edge sets $E^{\text{pos}\prime}$ and $E^{\text{neg}\prime}$

7       Update *layer*'s parameters using an optimizer of choice, *e.g.*, SGD and Adam

8      **end**

9      *best_layer* $\leftarrow$ load the layer with the best parameters found during the local training (lines 4 to 8)

10     $\boldsymbol{H} \leftarrow$ *best_layer*.forward($G$, $\boldsymbol{H}$)

11     $\boldsymbol{H} \leftarrow \boldsymbol{H}$.detach()

12   **end**

## D EXTENSIBILITY OF FORWARDGNN TO OTHER GRAPH LEARNING TASKS

In this work, we demonstrate how FORWARDGNN can be applied to the two fundamental graph learning tasks, namely, node classification and link prediction. Here we discuss the extensibility of FORWARDGNN to two other important graph learning tasks.

**Graph Classification.** Given a set of graphs, the goal of graph classification is to assign a label to each graph, *i.e.*, the prediction is done at the graph level as opposed to the node- and edge-level predictions made for node classification and link prediction, respectively. In the case of graph classification, GFF (Paliotta et al., 2023) presented the first FF-based approach. FORWARDGNN can be extended to improve upon GFF for graph classification by replacing the multiple forward passes required by GFF with a single forward pass. Similar to how FORWARDGNN learns to generate representatives of node classes to address node classification via a single forward pass, this extension requires effective mechanisms to learn class representatives at the graph level. To that end, we can investigate different approaches, *e.g.*, application of graph pooling methods (Liu et al., 2023), possibly in combination with virtual nodes, which can be used to identify and connect related nodes and graphs. Once we can generate class representatives of graph instances, the local contrastive learning objective used for node classification (Equation (6)) can be used similarly for graph classification.

**Graph Clustering.** Given a graph, the goal of graph clustering is to assign the nodes in the graph into (potentially overlapping) clusters. The major difference from node classification is that we do not have ground truth node labels for model training in graph clustering problems, and the number of clusters in the graph is usually not available. Due to this difference, the techniques presented in Section 3.1, including the use of virtual nodes, cannot be used for graph clustering. Thus, the learning signals (Equation (6)) designed for node classification cannot be directly utilized. Instead, the contrastive forward learning objective given by Equation (6) can be extended to incorporate the learning signals of deep graph clustering approaches (Liu et al., 2022), which can operate when node labels are unavailable and the number of classes are not known in advance. For instance, they construct self-supervised learning signals by utilizing input node features, and the characteristics of real-world graphs such as network homophily and hierarchical community structure (Park et al., 2022b). With the extended local learning objective, the proposed single-forward learning framework (Section 3.2) can be applied in nearly the same way for clustering the input graph.

## E ADDITIONAL RELATED WORK

In Sec. 2, we discuss alternatives to backpropagation (BP) for learning GNNs based on the forward-forward algorithm (Hinton, 2022). Here we present a review of other alternative approaches and related works for learning GNNs, as well as an overview of biologically-inspired learning algorithms.

### E.1 ALTERNATIVE APPROACHES AND RELATED WORK FOR LEARNING GNNS

Carreira-Perpiñán & Wang (2014) and Taylor et al. (2016) proposed alternative approaches to BP for learning neural networks (NNs), where the learning of NNs is cast as a constrained optimization problem in the Lagrangian framework, with neural computations expressed as constraints. These approaches are inherently local, in the sense that the gradient computation relies on neighboring neurons, not on the BP over the entire network. However, they are not scalable due to the high memory requirement, and thus are inapplicable to large-scale problems. Based on this Lagrangian-based approach, Tiezzi et al. (2020; 2022) presented an alternative method for learning GNNs. Specifically, they simplified the learning process of the GNN* model (Scarselli et al., 2009), a representative recurrent GNN model, which performs graph diffusion repeatedly until it converges to a stable fixed point. The LP-GNN proposed in Tiezzi et al. (2020; 2022) adopts a mixed strategy to enable a more efficient training of GNN*, which employs a constraint-based propagation scheme, thereby avoiding the explicit computation of the fixed point of the diffusion process in GNN*, while using BP in updating the NNs that model the state transition and output functions. Accordingly, although LP-GNN departs from the usual BP-based GNN training, it still relies on BP for learning GNNs. By contrast, FORWARDGNN is free from BP as a forward-only learning scheme for GNNs.

Stochastic steady-state embedding (SSE) (Dai et al., 2018) presents another learning scheme, in the context of learning steady-state solutions of iterative algorithms over graphs, *e.g.*, PageRank scores and connected components. To obtain steady-state solutions efficiently, SSE proposes a stochas-

tic fixed-point gradient descent process, inspired by the policy iteration in reinforcement learning. While SSE is an alternative learning approach, SSE is not, per se, a general training scheme for GNNs like FORWARDGNN, as its goal is to learn an algorithm achieving steady-state solutions of iterative graph algorithms, which differs from the task of GNN training FORWARDGNN addresses.

Another line of works investigated layer-wise approaches for learning GNNs. A layer-wise learning idea closely related to our work first appeared in the vision domain (Belilovsky et al., 2019), which proposed a greedy layer-wise learning technique for deep convolutional neural networks (CNNs), and obtained promising results on large-scale datasets such as ImageNet. In Belilovsky et al. (2019), a CNN model gets trained progressively and greedily, *i.e.*, layer by layer from the bottom to the top. To enable a layer-wise training, they formulate and solve an auxiliary problem for each layer, where an auxiliary classifier is attached to the layer, forming a block, and the original CNN layer and the auxiliary classifier in the block are jointly optimized in terms of the training accuracy of the auxiliary classifier. You et al. (2020) extended this approach for a layer-wise training of GNNs, in which the CNN layers are replaced with the GNN layers. Furthermore, You et al. (2020) streamlined the graph convolution operation by decoupling the feature aggregation and transformation, and introduced a learnable RNN controller, which automatically decides when to stop the training of each layer via reinforcement learning. A follow-up work by Wang et al. (2020) further improved the efficiency of layer-wise GNN training of You et al. (2020) via the proposed parallel training and lazy update scheme. While these methods train GNNs layer by layer, they still rely on BP, since each block is a multi-layer architecture in general, and as a result, the error derivatives are propagated across the layers in the block for optimization. By contrast, FORWARDGNN operates in the forward-only learning regime, which is the focus of this work. Also, their optimization framework allows only bottom-up signal propagation, and does not provide a way to incorporate the top-down signals. Thus, using the above layer-wise methods, the training of each layer cannot be informed by what the upper layers have learned as in FORWARDGNN. Further, they only investigate the node classification task, and miss out the other fundamental GL task of link prediction. In this work, we systematically explore the potential of forward-only learning for both node classification and link prediction tasks.

### E.2 BIOLOGICALLY-INSPIRED LEARNING ALGORITHMS

For the past several decades, many biologically-inspired algorithms have been developed to train neural networks without relying on the biologically implausible backpropagation of errors. A recent survey (Ororbia, 2023) organizes them based on how they address the central question of a learning algorithm, namely, *"Where do the signals for learning the elements of a neural network come from, and how they are produced?"* Following the taxonomy of Ororbia (2023), biologically-inspired algorithms can be grouped as follows.

**Implicit Signal Algorithms.** No explicit targets or signals are involved in these algorithms. Instead, they utilize implicit signals, which are produced by the feedforward process of the neural network. The computations for inference in these methods are the same as those for model training, and parameter updates are done using purely local information. For example, in Hebbian learning-based approaches (Journé et al., 2023; Gupta et al., 2021; Levy & Steward, 1983), only the pre-synaptic and post-synaptic activations are used to adjust the weights connecting the adjacent layers (*i.e.*, two-factor Hebbian), rendering the parameter adaptation to be a correlation-based learning.

**Global Explicit Signal Algorithms.** In algorithms in this and the following categories, the learning process is driven by explicit signals. Among them, approaches in this category adopt a global approach, in which a global feedback pathway is employed to carry signals for adjusting synaptic weights. Random feedback alignment and its variants (Lillicrap et al., 2016; Nøkland, 2016; Launay et al., 2019) construct teaching signals using random feedback weights, and establish asymmetric forward and backward pathways, thereby addressing the weight transport problem of BP, in which the same parameter matrices are used for both the inference and model training. Neuromodulatory approaches, such as three-factor Hebbian plasticity (Kuśmierz et al., 2017), are another family of algorithms in this category, which produce and broadcast modulatory signals that drive synaptic adjustments, *e.g.*, binary gating variables to either accept or reject a synaptic adjustment.

**Non-Synergistic Local Explicit Signal Algorithms.** Algorithms in this category adopt exclusively local machinery, such as a local classifier, to produce signals based on locally available information (*e.g.*, information from a pair of adjacent layers), which drive the learning of a network. One group of algorithms in this category can be referred to as synthetic local updates (SLU) (Jaderberg et al.,

2017; Czarnecki et al., 2017; Schmidhuber, 1990). The basic idea of SLU is to extend each layer with a learnable parametric model (*e.g.*, an additional weight matrix) to approximate and predict the weight updates or gradients based on the local information from adjacent layers. Local signalers (predictors) (Mostafa et al., 2018; Ma et al., 2023) present a different yet related approach based on non-synergistic signals, where each layer is augmented with a local predictor (*e.g.*, classifier), which projects the layer's activation into the class score distribution via a fixed random weight matrix, and gets adjusted according to the local cost.

**Synergistic Local Explicit Signal Algorithms.** In contrast to the non-synergistic schemes discussed above, algorithms in this category produce local learning signals via varying degrees of (indirect) coordination among the layers of a network. Note that this coordination across layers is not through a single global pathway as in the global explicit signal algorithms or BP. One group of algorithms in this class are based on discrepancy reduction, which performs synaptic updates by reducing the mismatch between the model's current representations and the the target representations produced by complementary neural processes, which may span several layers, *e.g.*, an approximate inversion pathway of the network in the case of target propagation approaches (Bengio, 2014; Lee et al., 2015; Meulemans et al., 2020), and synaptic message passing structure in the case of local representation alignment schemes (Ororbia & Mali, 2019; Ororbia et al., 2018).

Forward-only learning approaches (Hinton, 2022; Heinz, 1995; Kohan et al., 2023; 2018; Linsker, 1988; Zhao et al., 2023; Dellaferrera & Kreiman, 2022) are another family of algorithms in this category, which performs synaptic updates only using the inference process of a neural system, *i.e.*, without involving (recurrent) feedback mechanisms as in some of the aforementioned algorithm families. Forward-only approaches can be divided into two groups, one based on supervised context and the other based on self-supervised context. The first group of methods (Kohan et al., 2023; 2018; Zhao et al., 2023; Dellaferrera & Kreiman, 2022) presents a forward-only scheme that performs synaptic adjustments driven by signals arising in supervised context, *e.g.*, they learn the model such that the representations of an input and the corresponding context are closely located via a set of local loss functions. We notice that this idea is related to FORWARDGNN as both generate local learning signals for the inputs and contexts, although the way they are produced as well as the inference process greatly differs. Importantly, prior works are not applicable to graphs and GNNs, and provide no or limited mechanisms for incorporating top-down signals. The second group consists of self-supervised context-driven approaches (Hinton, 2022; Lee & Song, 2023; Ororbia & Mali, 2023; Paliotta et al., 2023), such as the forward-forward algorithm (FF) (Hinton, 2022), which aim to distinguish between positive samples (*i.e.*, items taken from the real data distribution) and negative samples (*i.e.*, incorrect, adversarially generated items). Training with these algorithms is performed so as to maximize the goodness for positive samples, while minimizing it for negative samples. With this objective, the loss of a neural network is defined to be the sum of local layer-wise loss functionals based on the goodness. Thus, the design of goodness and how negative samples are generated are key to the effectiveness of these approaches. These algorithms perform two forward passes on positive and negative samples, where the second forward pass replaces the usual backward pass of BP. Building upon FF and follow-up works, FORWARDGNN makes further improvements for an effective forward learning of GNNs by removing the need to design and generate negative inputs via its single-forward learning framework, and letting each layer learn from both bottom-up and top-down signals without relying on BP. Section 2.2 provides a more detailed discussion on FF (Hinton, 2022) as well as its extensions and applications.

For a comprehensive review and discussion of biologically-inspired learning algorithms, we refer the reader to Ororbia (2023).

# F ADDITIONAL EXPERIMENTAL RESULTS

## F.1 NODE CLASSIFICATION RESULTS

- Figure 5 presents the node classification accuracy (y-axis) vs. memory usage increase (x-axis) per graph neural networks.
- Table 3 presents the node classification performance of GCN, SAGE, and GAT models averaged over five runs on five real-world graphs, obtained with a different number of layers, and using different learning approaches.
- Table 4 presents the GPU memory usage (in MB) during the GNN training for node classification, observed with a different number of layers, and using different learning approaches.

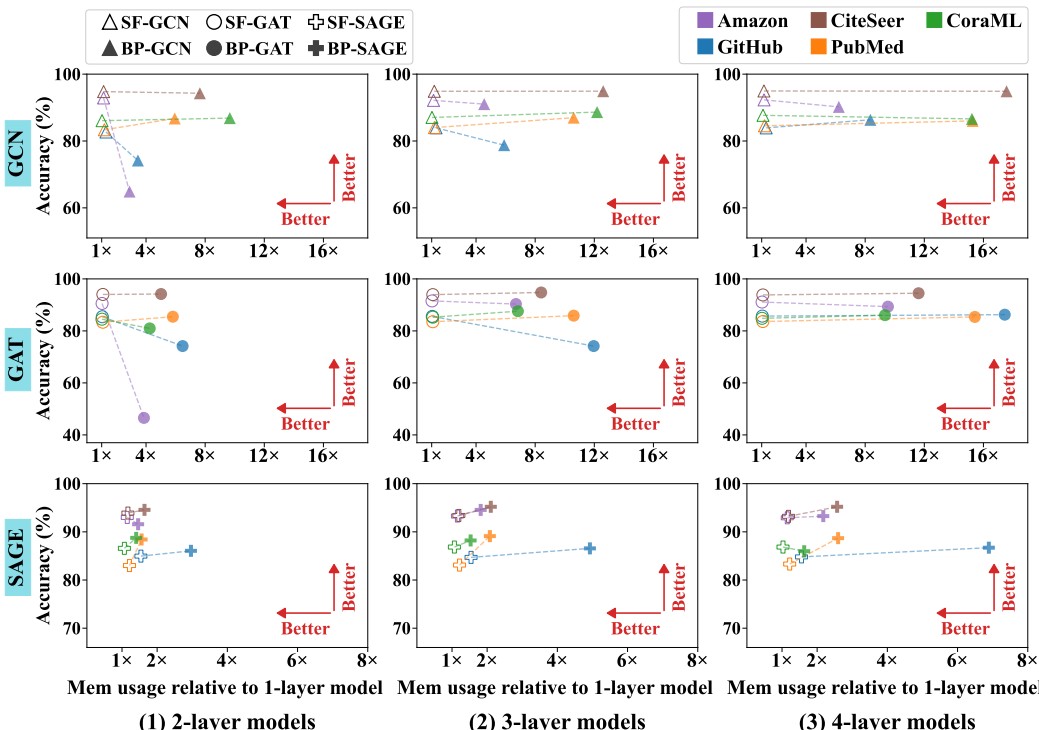

Figure 5: **Node classification** accuracy (y-axis) vs. memory usage increase (x-axis). The proposed single forward graph learning (SF, empty symbols) performs similarly to backpropagation (BP, filled symbols) in many cases (*i.e.*, near horizontal lines), or even better than BP in several cases (*i.e.*, lines going towards the lower right corner), with no increase in memory usage as more layers are used. Lines towards the lower right corner (SF outperforms BP), and upper right corner (BP outperforms SF); horizontal lines (both perform the same).

Table 3: Node classification performance of GCN, SAGE, and GAT averaged over five runs on five real-world graphs, obtained with a different number of layers, and using different learning methods. The best results are in **bold** font, and the best results among the forward learning methods are in red.

(a) AMAZON

| Method | Accuracy (↑) | | | |
|---|---|---|---|---|
| | # Layers=1 | # Layers=2 | # Layers=3 | # Layers=4 |
| Backpropagation-GCN | 45.83±0.4 | 64.88±16.1 | 91.03±1.3 | 90.21±2.2 |
| SymBa-GCN | 60.80±1.4 | 60.86±1.3 | 60.95±1.1 | 61.06±1.2 |
| CaFo-GCN | 42.52±1.2 | 61.02±8.9 | 60.82±8.9 | 60.69±9.4 |
| PEPITA-GCN | 13.87±6.8 | 21.82±4.9 | 19.80±3.5 | 15.57±4.0 |
| FF-LA-GCN (Sec 3.1) | 91.20±1.0 | 48.54±10.4 | 48.08±10.5 | 49.74±10.5 |
| FF-VN-GCN (Sec 3.1) | 22.69±1.6 | 26.43±4.5 | 22.92±3.8 | 20.84±3.3 |
| SF-GCN (Sec 3.2) | 93.67±0.6 | 93.73±0.4 | 93.48±0.2 | 93.48±0.3 |
| SF-Top-To-Loss-GCN (Sec 3.3) | 93.76±0.5 | 83.18±3.8 | 60.43±6.9 | 47.40±2.9 |
| SF-Top-To-Input-GCN (Sec 3.3) | 93.19±0.6 | 92.88±0.7 | 92.16±0.7 | 92.30±0.6 |
| Backpropagation-SAGE | 33.79±6.9 | 91.58±0.9 | **94.55**±0.4 | 93.27±0.4 |
| SymBa-SAGE | 91.87±1.1 | 91.87±0.9 | 91.75±0.9 | 91.71±0.9 |
| CaFo-SAGE | 35.86±5.6 | 36.07±5.8 | 36.16±5.8 | 33.80±6.3 |
| PEPITA-SAGE | 14.54±7.3 | 23.56±1.4 | 24.59±0.4 | 17.49±4.6 |
| FF-LA-SAGE (Sec 3.1) | 88.95±1.1 | 88.80±1.3 | 88.80±1.3 | 88.75±1.2 |
| FF-VN-SAGE (Sec 3.1) | 76.44±2.3 | 80.93±1.9 | 82.44±2.0 | 82.80±2.4 |
| SF-SAGE (Sec 3.2) | 92.90±0.3 | 93.63±0.4 | 93.70±0.4 | 93.71±0.3 |
| SF-Top-To-Loss-SAGE (Sec 3.3) | 93.01±0.8 | 87.88±2.4 | 87.62±1.9 | 84.92±1.8 |
| SF-Top-To-Input-SAGE (Sec 3.3) | 92.18±0.9 | 92.99±0.9 | 93.28±0.8 | 92.92±0.8 |
| Backpropagation-GAT | 37.63±5.4 | 46.54±16.1 | 90.33±1.7 | 89.33±0.7 |
| SymBa-GAT | **92.12**±0.5 | **92.30**±0.5 | **92.30**±0.5 | **92.31**±0.4 |
| CaFo-GAT | 44.88±1.6 | 45.33±1.5 | 50.86±11.8 | 52.10±10.8 |
| PEPITA-GAT | 17.19±6.1 | 16.27±7.1 | 22.80±7.4 | 18.94±5.9 |
| FF-LA-GAT (Sec 3.1) | 90.93±1.2 | 39.28±16.0 | 39.05±16.0 | 39.74±16.0 |
| FF-VN-GAT (Sec 3.1) | 47.65±28.5 | 49.40±27.9 | 50.20±28.3 | 50.16±28.6 |
| SF-GAT (Sec 3.2) | 60.07±1.4 | 87.02±0.7 | 90.18±0.6 | 90.68±0.8 |
| SF-Top-To-Loss-GAT (Sec 3.3) | 63.14±3.1 | 90.26±1.1 | 90.73±1.1 | 91.46±0.9 |
| SF-Top-To-Input-GAT (Sec 3.3) | 89.96±0.9 | 90.55±1.6 | 91.53±1.1 | 91.10±1.0 |

(b) GITHUB

| Method | Accuracy (↑) | | | |
|---|---|---|---|---|
| | # Layers=1 | # Layers=2 | # Layers=3 | # Layers=4 |
| Backpropagation-GCN | 74.16±0.5 | 74.17±0.5 | 78.76±6.0 | **86.34**±0.2 |
| SymBa-GCN | 74.16±0.5 | 74.19±0.5 | 74.46±0.3 | 74.47±0.4 |
| CaFo-GCN | 74.16±0.5 | 74.17±0.5 | 74.16±0.5 | 74.17±0.5 |
| PEPITA-GCN | 55.11±22.4 | 74.14±0.5 | 74.17±0.5 | 74.16±0.5 |
| FF-LA-GCN (Sec 3.1) | 80.90±0.5 | 77.19±1.7 | 75.69±1.1 | 74.91±0.5 |
| FF-VN-GCN (Sec 3.1) | 74.16±0.5 | 67.87±13.4 | 59.29±19.1 | 61.67±15.8 |
| SF-GCN (Sec 3.2) | 85.74±0.2 | 85.90±0.2 | 85.86±0.2 | 85.87±0.3 |
| SF-Top-To-Loss-GCN (Sec 3.3) | 85.91±0.2 | 76.56±1.5 | 74.17±0.5 | 74.17±0.5 |
| SF-Top-To-Input-GCN (Sec 3.3) | 80.66±0.4 | 82.69±1.0 | 83.94±0.6 | 83.92±0.5 |
| Backpropagation-SAGE | 74.17±0.5 | 86.06±0.3 | **86.57**±0.2 | **86.71**±0.4 |
| SymBa-SAGE | **86.44**±0.4 | **86.31**±0.5 | 86.37±0.4 | 86.45±0.4 |
| CaFo-SAGE | 74.16±0.5 | 74.17±0.5 | 74.17±0.5 | 74.17±0.5 |
| PEPITA-SAGE | 56.53±22.1 | 74.16±0.5 | 74.17±0.5 | 74.40±0.9 |
| FF-LA-SAGE (Sec 3.1) | 76.21±3.0 | 75.58±2.5 | 75.11±1.4 | 74.67±0.8 |
| FF-VN-SAGE (Sec 3.1) | 85.66±0.1 | 85.88±0.3 | 86.02±0.4 | 86.09±0.4 |
| SF-SAGE (Sec 3.2) | 80.27±1.4 | 81.96±0.9 | 82.92±0.7 | 83.54±0.6 |
| SF-Top-To-Loss-SAGE (Sec 3.3) | 79.80±0.8 | 82.15±1.5 | 82.27±1.3 | 83.03±0.8 |
| SF-Top-To-Input-SAGE (Sec 3.3) | 81.98±0.4 | 84.96±0.3 | 84.70±0.7 | 84.80±0.5 |
| Backpropagation-GAT | 74.17±0.5 | 74.17±0.5 | 74.17±0.5 | 86.24±0.2 |
| SymBa-GAT | **86.76**±0.4 | **86.67**±0.4 | 86.66±0.4 | 86.32±0.6 |
| CaFo-GAT | 74.16±0.5 | 74.17±0.5 | 74.17±0.5 | 74.17±0.5 |
| PEPITA-GAT | 63.50±18.4 | 75.26±2.1 | 74.20±0.5 | 74.22±0.6 |
| FF-LA-GAT (Sec 3.1) | 82.14±0.3 | 79.59±2.2 | 79.47±2.1 | 80.18±2.6 |
| FF-VN-GAT (Sec 3.1) | 85.97±0.3 | 86.29±0.3 | 86.70±0.3 | 86.64±0.3 |
| SF-GAT (Sec 3.2) | 74.29±0.7 | 83.01±0.8 | 83.83±0.6 | 84.71±0.6 |
| SF-Top-To-Loss-GAT (Sec 3.3) | 74.27±0.6 | 79.88±1.8 | 82.87±1.2 | 81.45±1.7 |
| SF-Top-To-Input-GAT (Sec 3.3) | 79.82±1.0 | 85.44±0.2 | 85.57±0.4 | 85.69±0.5 |

Table 3: *(Continued from the previous table)* Node classification performance of GCN, SAGE, and GAT averaged over five runs on five real-world graphs, obtained with a different number of layers, and using different learning methods. The best results are in **bold** font, and the best results among the forward learning methods are in red.

(c) CITESEER

| Method | Accuracy (↑) | | | |
| --- | --- | --- | --- | --- |
| | # Layers=1 | # Layers=2 | # Layers=3 | # Layers=4 |
| Backpropagation-GCN | 84.28±3.0 | 94.28±0.7 | 94.89±0.8 | 94.87±0.5 |
| SymBa-GCN | 87.33±1.5 | 88.06±1.3 | 88.01±1.4 | 88.01±1.4 |
| CaFo-GCN | 68.25±2.2 | 69.79±0.5 | 69.31±0.7 | 69.29±1.3 |
| PEPITA-GCN | 19.22±4.5 | 19.53±3.4 | 17.09±1.7 | 19.31±2.1 |
| FF-LA-GCN (Sec 3.1) | **95.39**±0.4 | **95.86**±0.7 | **95.89**±0.6 | **95.89**±0.6 |
| FF-VN-GCN (Sec 3.1) | 43.22±14.2 | 53.31±10.7 | 52.81±10.4 | 54.28±11.2 |
| SF-GCN (Sec 3.2) | 92.60±0.7 | 94.18±0.7 | 94.33±0.7 | 94.66±0.6 |
| SF-Top-To-Loss-GCN (Sec 3.3) | 92.62±0.4 | 94.11±0.5 | 94.56±0.5 | 94.02±0.7 |
| SF-Top-To-Input-GCN (Sec 3.3) | 94.30±0.4 | 94.78±0.2 | 94.87±0.5 | 94.96±0.7 |
| Backpropagation-SAGE | 87.28±0.9 | 94.56±0.8 | 95.20±0.7 | 95.20±0.7 |
| SymBa-SAGE | 88.53±0.9 | 91.99±1.6 | 93.85±0.4 | 93.76±0.5 |
| CaFo-SAGE | 61.70±2.0 | 61.68±2.0 | 60.99±2.2 | 61.02±3.2 |
| PEPITA-SAGE | 20.45±4.0 | 21.80±4.2 | 19.03±3.1 | 18.13±1.7 |
| FF-LA-SAGE (Sec 3.1) | **95.41**±0.7 | **95.53**±0.6 | **95.58**±0.8 | **95.63**±0.7 |
| FF-VN-SAGE (Sec 3.1) | 89.95±1.0 | 93.85±0.6 | 95.39±0.6 | 95.56±0.7 |
| SF-SAGE (Sec 3.2) | 79.88±2.1 | 89.60±1.8 | 92.27±1.1 | 92.74±1.0 |
| SF-Top-To-Loss-SAGE (Sec 3.3) | 79.39±1.7 | 92.51±0.7 | 94.02±0.8 | 93.43±0.9 |
| SF-Top-To-Input-SAGE (Sec 3.3) | 90.57±0.5 | 93.90±0.5 | 93.40±0.7 | 93.22±0.7 |
| Backpropagation-GAT | 79.17±1.5 | 94.18±0.7 | 94.78±1.0 | 94.49±0.8 |
| SymBa-GAT | 91.51±0.8 | 95.01±1.1 | 95.25±0.9 | 95.18±0.9 |
| CaFo-GAT | 73.29±0.7 | 72.60±1.0 | 72.34±0.7 | 71.51±0.8 |
| PEPITA-GAT | 15.65±2.8 | 16.93±1.5 | 17.94±3.2 | 18.79±3.3 |
| FF-LA-GAT (Sec 3.1) | **95.34**±0.4 | **95.48**±0.6 | 95.27±0.5 | 95.37±0.6 |
| FF-VN-GAT (Sec 3.1) | 90.99±1.2 | 94.04±1.0 | **95.48**±0.6 | **95.39**±0.6 |
| SF-GAT (Sec 3.2) | 82.06±2.4 | 93.33±0.9 | 93.78±1.1 | 94.21±1.0 |
| SF-Top-To-Loss-GAT (Sec 3.3) | 81.21±2.6 | 91.96±0.8 | 92.03±0.7 | 91.89±0.8 |
| SF-Top-To-Input-GAT (Sec 3.3) | 91.16±0.8 | 94.02±0.7 | 93.95±0.8 | 93.83±0.6 |

(d) PUBMED

| Method | Accuracy (↑) | | | |
| --- | --- | --- | --- | --- |
| | # Layers=1 | # Layers=2 | # Layers=3 | # Layers=4 |
| Backpropagation-GCN | 63.82±1.0 | 86.74±0.3 | 86.98±0.7 | 86.01±0.5 |
| SymBa-GCN | 82.13±0.8 | 82.10±0.7 | 82.06±0.7 | 82.04±0.7 |
| CaFo-GCN | 69.90±1.4 | 69.16±2.1 | 69.78±2.0 | 68.89±1.5 |
| PEPITA-GCN | 33.35±4.6 | 39.66±4.7 | 42.17±2.4 | 41.91±2.7 |
| FF-LA-GCN (Sec 3.1) | 81.33±0.4 | 83.57±0.4 | 83.88±0.3 | 83.97±0.3 |
| FF-VN-GCN (Sec 3.1) | 51.69±1.2 | 52.89±1.5 | 52.96±1.1 | 54.07±4.0 |
| SF-GCN (Sec 3.2) | **86.98**±0.2 | **87.61**±0.3 | **87.66**±0.4 | **87.61**±0.4 |
| SF-Top-To-Loss-GCN (Sec 3.3) | 86.82±0.5 | 86.59±0.6 | 86.17±0.3 | 85.69±0.8 |
| SF-Top-To-Input-GCN (Sec 3.3) | 77.91±0.9 | 83.36±0.7 | 84.01±0.7 | 84.55±0.5 |
| Backpropagation-SAGE | 70.66±3.3 | 88.42±0.5 | **89.11**±0.4 | 88.69±0.4 |
| SymBa-SAGE | 88.27±0.6 | 88.33±0.5 | 88.44±0.4 | 88.30±0.3 |
| CaFo-SAGE | 62.66±2.8 | 63.69±3.1 | 63.01±3.3 | 63.34±3.2 |
| PEPITA-SAGE | 33.24±5.3 | 40.19±4.8 | 41.20±1.7 | 40.62±2.0 |
| FF-LA-SAGE (Sec 3.1) | 83.54±0.5 | 83.48±0.6 | 83.54±0.6 | 83.50±0.6 |
| FF-VN-SAGE (Sec 3.1) | **88.83**±0.5 | **88.93**±0.5 | 88.95±0.5 | **88.89**±0.4 |
| SF-SAGE (Sec 3.2) | 78.49±1.3 | 81.33±0.7 | 82.87±0.7 | 83.54±0.8 |
| SF-Top-To-Loss-SAGE (Sec 3.3) | 78.23±1.0 | 80.84±0.7 | 81.95±0.6 | 82.07±0.4 |
| SF-Top-To-Input-SAGE (Sec 3.3) | 79.53±0.5 | 83.00±0.6 | 83.11±0.5 | 83.32±0.5 |
| Backpropagation-GAT | 63.68±0.5 | 85.45±0.4 | 85.86±0.6 | 85.37±0.6 |
| SymBa-GAT | **88.93**±0.4 | **89.47**±0.2 | **89.60**±0.4 | **89.53**±0.5 |
| CaFo-GAT | 73.23±0.9 | 73.03±0.8 | 71.85±1.4 | 71.61±1.6 |
| PEPITA-GAT | 32.23±8.4 | 35.69±7.0 | 41.99±2.2 | 39.28±1.7 |
| FF-LA-GAT (Sec 3.1) | 81.62±0.4 | 83.55±0.6 | 83.40±0.4 | 83.43±0.4 |
| FF-VN-GAT (Sec 3.1) | 88.22±0.4 | 88.77±0.6 | 88.86±0.6 | 88.86±0.6 |
| SF-GAT (Sec 3.2) | 64.41±7.4 | 78.41±2.6 | 81.34±1.5 | 82.57±1.1 |
| SF-Top-To-Loss-GAT (Sec 3.3) | 63.64±6.5 | 79.11±1.5 | 81.71±0.9 | 82.15±0.9 |
| SF-Top-To-Input-GAT (Sec 3.3) | 77.24±4.1 | 83.34±0.7 | 83.53±0.5 | 83.61±0.4 |

Table 3: *(Continued from the previous table)* Node classification performance of GCN, SAGE, and GAT averaged over five runs on five real-world graphs, obtained with a different number of layers, and using different learning methods. The best results are in **bold** font, and the best results among the forward learning methods are in red.

(e) CORAML

| Method | Accuracy (↑) | | | |
|---|---|---|---|---|
| | # Layers=1 | # Layers=2 | # Layers=3 | # Layers=4 |
| Backpropagation-GCN | 31.72±4.8 | 86.84±1.0 | **88.65**±1.4 | 86.61±2.3 |
| SymBa-GCN | 82.94±1.7 | 82.97±1.7 | 83.37±1.8 | 83.34±1.9 |
| CaFo-GCN | 31.45±2.8 | 30.28±4.1 | 29.65±4.0 | 28.95±2.4 |
| PEPITA-GCN | 13.19±1.9 | 16.76±5.8 | 15.63±2.0 | 18.30±8.0 |
| FF-LA-GCN (Sec 3.1) | 81.74±1.9 | 63.77±2.7 | 63.71±2.7 | 64.54±3.0 |
| FF-VN-GCN (Sec 3.1) | 28.61±1.9 | 28.11±3.2 | 27.25±3.5 | 25.81±4.6 |
| SF-GCN (Sec 3.2) | 87.75±1.4 | 87.95±1.4 | 88.15±1.5 | 88.48±1.2 |
| SF-Top-To-Loss-GCN (Sec 3.3) | 87.55±1.4 | 84.07±0.9 | 85.64±1.6 | 84.67±2.6 |
| SF-Top-To-Input-GCN (Sec 3.3) | 87.05±1.4 | 86.08±1.6 | 87.05±1.5 | 87.68±1.3 |
| Backpropagation-SAGE | 42.87±5.0 | **88.75**±1.1 | **88.21**±1.8 | 85.98±2.2 |
| SymBa-SAGE | 87.45±0.8 | 87.28±0.7 | 87.21±0.7 | 86.98±1.0 |
| CaFo-SAGE | 28.98±2.1 | 28.71±2.1 | 28.58±1.9 | 28.95±1.6 |
| PEPITA-SAGE | 13.39±1.2 | 25.31±4.4 | 17.13±5.2 | 19.13±8.9 |
| FF-LA-SAGE (Sec 3.1) | 85.41±1.4 | 85.34±0.8 | 85.08±1.3 | 85.18±1.3 |
| FF-VN-SAGE (Sec 3.1) | 86.01±0.7 | 87.68±1.6 | 87.85±1.4 | 87.71±1.3 |
| SF-SAGE (Sec 3.2) | 85.14±1.6 | 87.35±1.4 | 87.61±1.5 | 87.55±1.5 |
| SF-Top-To-Loss-SAGE (Sec 3.3) | 84.77±1.4 | 85.94±1.3 | 85.28±1.3 | 85.31±1.8 |
| SF-Top-To-Input-SAGE (Sec 3.3) | 86.08±2.4 | 86.58±1.2 | 86.84±1.2 | 86.84±1.1 |
| Backpropagation-GAT | 33.82±2.1 | 80.97±1.1 | 87.58±1.6 | 86.04±1.7 |
| SymBa-GAT | 86.68±1.3 | 87.51±1.2 | 87.68±0.7 | 87.38±0.6 |
| CaFo-GAT | 30.98±3.7 | 32.95±4.9 | 37.96±15.8 | 36.29±16.9 |
| PEPITA-GAT | 15.13±4.3 | 22.54±6.6 | 15.13±2.7 | 16.66±6.7 |
| FF-LA-GAT (Sec 3.1) | 82.67±1.2 | 61.80±6.4 | 63.57±4.9 | 63.24±4.1 |
| FF-VN-GAT (Sec 3.1) | 84.61±1.5 | 85.64±1.5 | 86.48±2.1 | 86.74±2.1 |
| SF-GAT (Sec 3.2) | 66.81±2.0 | 83.17±2.0 | 84.34±1.9 | 85.14±2.2 |
| SF-Top-To-Loss-GAT (Sec 3.3) | 65.21±2.9 | 79.50±2.4 | 83.57±1.5 | 84.17±1.8 |
| SF-Top-To-Input-GAT (Sec 3.3) | 83.27±2.0 | 84.51±2.1 | 85.28±1.7 | 84.84±2.8 |

Table 4: GPU memory usage (in MB) for node classification. The best results are in **bold** font, and the best results among the forward learning methods are in red.

(a) CITESEER

| Method | GPU Memory Usage (↓) | | | |
|---|---|---|---|---|
| | # Layers=1 | # Layers=2 | # Layers=3 | # Layers=4 |
| Backpropagation-GCN | **0.52** | 4.01 | 6.61 | 9.21 |
| SymBa-GCN | 30.56 | 30.62 | 30.62 | 30.62 |
| CaFo-GCN | 3.40 | 3.40 | 3.40 | 3.40 |
| PEPITA-GCN | 16.56 | 22.04 | 25.28 | 29.70 |
| FF-LA-GCN (Sec 3.1) | 24.50 | 25.20 | 25.20 | 25.20 |
| FF-VN-GCN (Sec 3.1) | 25.23 | 25.29 | 25.29 | 25.29 |
| SF-GCN (Sec 3.2) | 15.51 | 15.51 | 15.51 | 15.51 |
| SF-Top-To-Loss-GCN (Sec 3.3) | 11.10 | 13.17 | 15.24 | 17.11 |
| SF-Top-To-Input-GCN (Sec 3.3) | 16.62 | 18.79 | 19.53 | 19.12 |
| Backpropagation-SAGE | 10.01 | 16.42 | 21.07 | 25.72 |
| SymBa-SAGE | 89.40 | 89.40 | 89.40 | 89.40 |
| CaFo-SAGE | 4.58 | 4.58 | 4.58 | 4.58 |
| PEPITA-SAGE | 25.99 | 34.26 | 40.29 | 46.54 |
| FF-LA-SAGE (Sec 3.1) | 84.54 | 84.54 | 84.54 | 84.54 |
| FF-VN-SAGE (Sec 3.1) | 83.45 | 83.45 | 83.45 | 83.45 |
| SF-SAGE (Sec 3.2) | 26.03 | 26.03 | 26.03 | 26.03 |
| SF-Top-To-Loss-SAGE (Sec 3.3) | 23.34 | 25.40 | 26.93 | 29.00 |
| SF-Top-To-Input-SAGE (Sec 3.3) | 28.30 | 33.05 | 33.79 | 33.52 |
| Backpropagation-GAT | 3.81 | 19.14 | 32.07 | 44.30 |
| SymBa-GAT | 111.56 | 111.87 | 111.87 | 111.87 |
| CaFo-GAT | 3.41 | 3.41 | 3.41 | 3.41 |
| PEPITA-GAT | 17.20 | 32.33 | 46.58 | 63.22 |
| FF-LA-GAT (Sec 3.1) | 88.15 | 88.18 | 88.60 | 88.60 |
| FF-VN-GAT (Sec 3.1) | 106.06 | 106.54 | 106.54 | 106.54 |
| SF-GAT (Sec 3.2) | 28.13 | 28.13 | 28.13 | 28.13 |
| SF-Top-To-Loss-GAT (Sec 3.3) | 24.39 | 26.27 | 28.53 | 30.60 |
| SF-Top-To-Input-GAT (Sec 3.3) | 30.16 | 32.56 | 32.46 | 32.56 |

Table 4: *(Continued from the previous table)* GPU memory usage (in MB) for node classification. The best results are in **bold** font, and the best results among the forward learning methods are in red.

(b) PUBMED

| Method | GPU Memory Usage (↓) | | | |
|---|---|---|---|---|
| | # Layers=1 | # Layers=2 | # Layers=3 | # Layers=4 |
| Backpropagation-GCN | **2.56** | 15.22 | 27.16 | 39.11 |
| SymBa-GCN | 62.58 | 63.22 | 63.93 | 63.96 |
| CaFo-GCN | 11.00 | 11.00 | 11.00 | 11.00 |
| PEPITA-GCN | 64.82 | 84.82 | 102.92 | 121.62 |
| FF-LA-GCN (Sec 3.1) | 55.26 | 55.26 | 55.26 | 55.26 |
| FF-VN-GCN (Sec 3.1) | 56.37 | 56.37 | 57.72 | 57.75 |
| SF-GCN (Sec 3.2) | 57.72 | 57.72 | 57.72 | 57.72 |
| SF-Top-To-Loss-GCN (Sec 3.3) | 41.14 | 50.77 | 60.39 | 69.59 |
| SF-Top-To-Input-GCN (Sec 3.3) | 58.71 | 69.43 | 69.72 | 69.95 |
| Backpropagation-SAGE | 38.12 | 59.38 | 79.21 | 99.04 |
| SymBa-SAGE | 168.62 | 168.62 | 168.62 | 168.62 |
| CaFo-SAGE | 11.98 | 11.98 | 11.98 | 11.98 |
| PEPITA-SAGE | 99.97 | 126.34 | 152.01 | 177.67 |
| FF-LA-SAGE (Sec 3.1) | 163.08 | 163.08 | 163.08 | 163.08 |
| FF-VN-SAGE (Sec 3.1) | 162.41 | 162.41 | 162.41 | 162.41 |
| SF-SAGE (Sec 3.2) | 93.77 | 93.77 | 93.77 | 93.77 |
| SF-Top-To-Loss-SAGE (Sec 3.3) | 76.98 | 86.61 | 96.24 | 105.87 |
| SF-Top-To-Input-SAGE (Sec 3.3) | 95.97 | 116.35 | 116.58 | 116.87 |
| Backpropagation-GAT | 16.96 | 98.61 | 179.90 | 261.51 |
| SymBa-GAT | 316.04 | 316.04 | 316.04 | 316.04 |
| CaFo-GAT | 11.00 | 11.75 | 11.75 | 11.75 |
| PEPITA-GAT | 68.37 | 156.77 | 243.17 | 333.78 |
| FF-LA-GAT (Sec 3.1) | 265.23 | 265.23 | 265.25 | 265.25 |
| FF-VN-GAT (Sec 3.1) | 309.59 | 309.62 | 309.62 | 309.62 |
| SF-GAT (Sec 3.2) | 141.61 | 141.61 | 141.61 | 141.61 |
| SF-Top-To-Loss-GAT (Sec 3.3) | 125.81 | 135.07 | 145.07 | 154.15 |
| SF-Top-To-Input-GAT (Sec 3.3) | 143.29 | 154.68 | 154.22 | 154.38 |

(c) AMAZON

| Method | GPU Memory Usage (↓) | | | |
|---|---|---|---|---|
| | # Layers=1 | # Layers=2 | # Layers=3 | # Layers=4 |
| Backpropagation-GCN | **5.20** | 15.01 | 23.69 | 32.36 |
| SymBa-GCN | 106.78 | 106.78 | 106.78 | 106.78 |
| CaFo-GCN | 5.55 | 5.55 | 5.56 | 5.56 |
| PEPITA-GCN | 41.12 | 53.31 | 64.39 | 75.46 |
| FF-LA-GCN (Sec 3.1) | 88.53 | 89.43 | 89.43 | 89.43 |
| FF-VN-GCN (Sec 3.1) | 92.33 | 92.33 | 92.33 | 92.33 |
| SF-GCN (Sec 3.2) | 34.81 | 34.81 | 34.81 | 34.81 |
| SF-Top-To-Loss-GCN (Sec 3.3) | 21.79 | 26.08 | 29.27 | 34.11 |
| SF-Top-To-Input-GCN (Sec 3.3) | 36.30 | 40.89 | 40.77 | 42.11 |
| Backpropagation-SAGE | 22.35 | 32.61 | 40.61 | 48.61 |
| SymBa-SAGE | 243.25 | 243.25 | 243.25 | 243.25 |
| CaFo-SAGE | 7.00 | 7.00 | 7.00 | 7.00 |
| PEPITA-SAGE | 58.43 | 69.47 | 79.58 | 89.69 |
| FF-LA-SAGE (Sec 3.1) | 230.49 | 230.49 | 230.49 | 230.49 |
| FF-VN-SAGE (Sec 3.1) | 228.54 | 228.54 | 228.54 | 228.54 |
| SF-SAGE (Sec 3.2) | 53.15 | 53.15 | 53.15 | 53.15 |
| SF-Top-To-Loss-SAGE (Sec 3.3) | 41.54 | 45.28 | 49.02 | 53.68 |
| SF-Top-To-Input-SAGE (Sec 3.3) | 56.36 | 64.81 | 65.04 | 64.81 |
| Backpropagation-GAT | 51.66 | 199.31 | 346.07 | 492.84 |
| SymBa-GAT | 1255.72 | 1255.72 | 1255.72 | 1255.72 |
| CaFo-GAT | 5.55 | 5.55 | 5.55 | 5.55 |
| PEPITA-GAT | 51.99 | 201.99 | 351.13 | 500.44 |
| FF-LA-GAT (Sec 3.1) | 1193.36 | 1193.36 | 1193.36 | 1193.36 |
| FF-VN-GAT (Sec 3.1) | 1241.26 | 1241.26 | 1241.26 | 1241.26 |
| SF-GAT (Sec 3.2) | 178.39 | 178.39 | 178.39 | 178.39 |
| SF-Top-To-Loss-GAT (Sec 3.3) | 166.81 | 170.87 | 174.34 | 178.02 |
| SF-Top-To-Input-GAT (Sec 3.3) | 181.03 | 185.83 | 185.01 | 185.83 |

Table 4: *(Continued from the previous table)* GPU memory usage (in MB) for node classification. The best results are in **bold** font, and the best results among the forward learning methods are in red.

(d) GITHUB

| Method | GPU Memory Usage (↓) | | | |
|---|---|---|---|---|
| | # Layers=1 | # Layers=2 | # Layers=3 | # Layers=4 |
| Backpropagation-GCN | **12.41** | 42.81 | 73.22 | 103.62 |
| SymBa-GCN | 86.64 | 87.01 | 87.01 | 87.01 |
| CaFo-GCN | 19.22 | **19.22** | **19.22** | **19.22** |
| PEPITA-GCN | 42.78 | 84.96 | 127.15 | 169.33 |
| FF-LA-GCN (Sec 3.1) | 84.49 | 84.49 | 84.49 | 84.49 |
| FF-VN-GCN (Sec 3.1) | 86.65 | 87.01 | 87.01 | 87.01 |
| SF-GCN (Sec 3.2) | 62.64 | 62.64 | 62.64 | 62.64 |
| SF-Top-To-Loss-GCN (Sec 3.3) | 81.30 | 99.71 | 118.12 | 136.00 |
| SF-Top-To-Input-GCN (Sec 3.3) | 63.19 | 81.49 | 81.80 | 82.38 |
| Backpropagation-SAGE | **19.08** | 56.54 | 94.00 | 131.46 |
| SymBa-SAGE | 98.07 | 98.07 | 98.07 | 98.07 |
| CaFo-SAGE | 19.47 | **19.47** | **19.47** | **19.47** |
| PEPITA-SAGE | 49.44 | 98.29 | 147.14 | 195.98 |
| FF-LA-SAGE (Sec 3.1) | 98.65 | 98.65 | 98.65 | 98.65 |
| FF-VN-SAGE (Sec 3.1) | 98.07 | 98.07 | 98.07 | 98.07 |
| SF-SAGE (Sec 3.2) | 68.10 | 68.10 | 68.10 | 68.10 |
| SF-Top-To-Loss-SAGE (Sec 3.3) | 87.01 | 105.42 | 123.83 | 142.24 |
| SF-Top-To-Input-SAGE (Sec 3.3) | 69.19 | 106.42 | 106.71 | 107.57 |
| Backpropagation-GAT | 70.46 | 456.12 | 842.55 | 1228.98 |
| SymBa-GAT | 850.82 | 850.82 | 850.82 | 850.82 |
| CaFo-GAT | **19.22** | **19.22** | **19.22** | **19.22** |
| PEPITA-GAT | 55.10 | 453.16 | 851.37 | 1250.04 |
| FF-LA-GAT (Sec 3.1) | 796.80 | 797.53 | 797.53 | 797.53 |
| FF-VN-GAT (Sec 3.1) | 850.83 | 850.83 | 850.83 | 850.83 |
| SF-GAT (Sec 3.2) | 444.83 | 444.83 | 444.83 | 444.83 |
| SF-Top-To-Loss-GAT (Sec 3.3) | 463.50 | 481.91 | 500.32 | 519.08 |
| SF-Top-To-Input-GAT (Sec 3.3) | 445.10 | 463.59 | 463.59 | 463.59 |

(e) CORAML

| Method | GPU Memory Usage (↓) | | | |
|---|---|---|---|---|
| | # Layers=1 | # Layers=2 | # Layers=3 | # Layers=4 |
| Backpropagation-GCN | **0.83** | 7.99 | 10.07 | 12.58 |
| SymBa-GCN | 30.71 | 30.71 | 30.71 | 30.71 |
| CaFo-GCN | 7.22 | **7.61** | **7.61** | **7.61** |
| PEPITA-GCN | 55.80 | 63.90 | 66.92 | 69.93 |
| FF-LA-GCN (Sec 3.1) | 25.73 | 25.74 | 25.74 | 25.74 |
| FF-VN-GCN (Sec 3.1) | 25.99 | 26.14 | 26.14 | 26.14 |
| SF-GCN (Sec 3.2) | 41.56 | 41.56 | 41.56 | 41.56 |
| SF-Top-To-Loss-GCN (Sec 3.3) | 19.66 | 19.25 | 20.56 | 22.24 |
| SF-Top-To-Input-GCN (Sec 3.3) | 48.25 | 49.47 | 49.16 | 52.56 |
| Backpropagation-SAGE | 33.66 | 47.25 | 51.49 | 54.93 |
| SymBa-SAGE | 263.63 | 263.63 | 263.63 | 263.63 |
| CaFo-SAGE | **12.84** | **12.84** | **12.84** | **12.84** |
| PEPITA-SAGE | 87.37 | 96.56 | 100.67 | 105.23 |
| FF-LA-SAGE (Sec 3.1) | 259.62 | 259.62 | 259.62 | 259.62 |
| FF-VN-SAGE (Sec 3.1) | 258.92 | 258.92 | 258.92 | 258.92 |
| SF-SAGE (Sec 3.2) | 79.71 | 79.71 | 79.71 | 79.71 |
| SF-Top-To-Loss-SAGE (Sec 3.3) | 61.16 | 62.87 | 64.64 | 66.04 |
| SF-Top-To-Input-SAGE (Sec 3.3) | 92.35 | 98.58 | 98.57 | 94.90 |
| Backpropagation-GAT | **5.65** | 24.03 | 38.63 | 52.80 |
| SymBa-GAT | 134.08 | 134.08 | 134.08 | 134.08 |
| CaFo-GAT | 7.23 | **7.62** | **7.62** | **7.62** |
| PEPITA-GAT | 56.65 | 76.84 | 92.39 | 107.50 |
| FF-LA-GAT (Sec 3.1) | 110.94 | 111.28 | 111.28 | 111.28 |
| FF-VN-GAT (Sec 3.1) | 129.36 | 129.36 | 129.36 | 129.36 |
| SF-GAT (Sec 3.2) | 55.75 | 55.75 | 55.75 | 55.75 |
| SF-Top-To-Loss-GAT (Sec 3.3) | 34.49 | 33.53 | 34.91 | 38.37 |
| SF-Top-To-Input-GAT (Sec 3.3) | 62.67 | 63.76 | 67.17 | 64.20 |

## F.2 LINK PREDICTION RESULTS

- Figure 6 presents the link prediction accuracy (y-axis) vs. memory usage increase (x-axis) per graph neural networks.
- Table 5 presents the link prediction performance of GCN, SAGE, and GAT models averaged over five runs on five real-world graphs, obtained with a different number of layers, and using different learning approaches.
- Table 6 presents the GPU memory usage (in MB) during the GNN training for link prediction, observed with a different number of layers, and using different learning approaches.

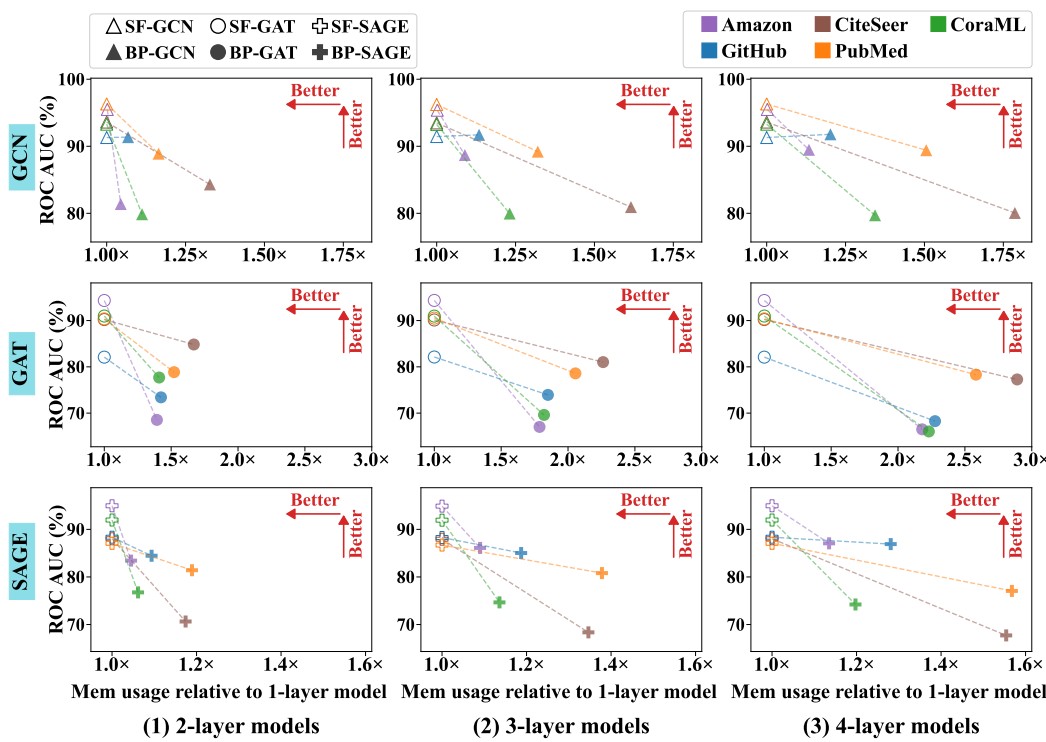

Figure 6: **Link prediction** performance (y-axis) vs. memory usage increase (x-axis). The proposed single-forward approach (SF, empty symbols) outperforms backpropagation (BP, filled symbols) (corresponding to most lines going towards the lower right corner), with no increase in memory usage as more layers are used. Lines towards the lower right corner (SF outperforms BP), and upper right corner (BP outperforms SF); horizontal lines (both perform the same).

Table 5: Link prediction performance of GCN, SAGE, and GAT averaged over five runs on five real-world graphs, obtained with a different number of layers, and using different learning methods. The best results are in **bold** font, and the best results among the forward learning methods are in red.

(a) CITESEER

| Method | ROC AUC (↑) | | | |
| --- | --- | --- | --- | --- |
| | # Layers=1 | # Layers=2 | # Layers=3 | # Layers=4 |
| Backpropagation-GCN | 90.00±0.6 | 84.31±0.6 | 80.93±1.0 | 80.05±0.9 |
| CaFo-GCN | 81.35±0.9 | 86.89±0.7 | 87.60±0.7 | 88.17±0.7 |
| ForwardGNN-SymBa-GCN | 89.83±0.3 | 90.65±1.1 | 90.09±0.9 | 90.65±1.1 |
| ForwardGNN-FF-GCN | 85.11±0.7 | 91.94±0.5 | 91.95±0.5 | 91.94±0.5 |
| ForwardGNN-CE-GCN | 89.32±2.4 | **93.61**±1.0 | **93.49**±0.9 | **93.61**±1.0 |
| ForwardGNN-CE-Top-To-Input-GCN | **92.15**±0.6 | 88.39±1.1 | 88.47±0.7 | 88.06±0.6 |
| Backpropagation-SAGE | 86.46±0.5 | 70.63±0.8 | 68.35±1.4 | 67.71±1.5 |
| CaFo-SAGE | 73.42±1.1 | 65.64±4.0 | 60.20±2.0 | 57.99±1.6 |
| ForwardGNN-SymBa-SAGE | 73.68±0.9 | 71.31±3.0 | 71.34±3.0 | 71.31±3.0 |
| ForwardGNN-FF-SAGE | 85.31±0.4 | 85.01±0.4 | 85.13±0.4 | 85.01±0.4 |
| ForwardGNN-CE-SAGE | **87.16**±0.6 | **88.02**±0.5 | **87.94**±0.5 | **88.02**±0.5 |
| ForwardGNN-CE-Top-To-Input-SAGE | 75.51±1.0 | 79.31±0.5 | 80.15±0.9 | 79.28±0.3 |
| Backpropagation-GAT | **88.42**±0.5 | 84.83±0.8 | 81.02±3.0 | 77.27±0.9 |
| CaFo-GAT | 69.67±1.5 | 84.61±1.2 | 85.15±1.0 | 85.05±1.0 |
| ForwardGNN-SymBa-GAT | 85.16±0.4 | 86.49±0.4 | 86.49±0.4 | 86.49±0.4 |
| ForwardGNN-FF-GAT | 78.14±5.5 | 89.64±0.7 | 89.62±0.7 | 89.64±0.7 |
| ForwardGNN-CE-GAT | 87.68±0.4 | **90.22**±0.6 | **90.12**±0.5 | **90.22**±0.6 |
| ForwardGNN-CE-Top-To-Input-GAT | 88.21±0.7 | 84.98±0.7 | 84.20±1.0 | 85.06±0.4 |

(b) PUBMED

| Method | ROC AUC (↑) | | | |
| --- | --- | --- | --- | --- |
| | # Layers=1 | # Layers=2 | # Layers=3 | # Layers=4 |
| Backpropagation-GCN | 91.83±0.3 | 88.90±0.2 | 89.20±0.2 | 89.41±0.2 |
| CaFo-GCN | 86.35±1.7 | 89.92±1.2 | 90.70±1.2 | 91.08±1.0 |
| ForwardGNN-SymBa-GCN | 96.20±0.2 | 95.56±0.5 | 95.74±0.3 | 95.56±0.5 |
| ForwardGNN-FF-GCN | 96.53±0.1 | **96.86**±0.1 | **96.87**±0.1 | **96.86**±0.1 |
| ForwardGNN-CE-GCN | 95.30±0.1 | 96.31±0.1 | 96.23±0.1 | 96.31±0.1 |
| ForwardGNN-CE-Top-To-Input-GCN | **96.84**±0.1 | 94.15±0.3 | 94.03±0.2 | 93.96±0.2 |
| Backpropagation-SAGE | 59.38±0.8 | 81.41±0.5 | 80.81±1.0 | 77.06±2.4 |
| CaFo-SAGE | 73.47±1.0 | 68.17±1.4 | 66.22±1.5 | 66.95±1.0 |
| ForwardGNN-SymBa-SAGE | 77.61±0.2 | 77.56±0.2 | 77.56±0.2 | 77.56±0.2 |
| ForwardGNN-FF-SAGE | 82.92±0.5 | 82.69±0.5 | 82.81±0.5 | 82.69±0.5 |
| ForwardGNN-CE-SAGE | 83.81±0.4 | **87.07**±0.4 | **86.71**±0.4 | **87.07**±0.4 |
| ForwardGNN-CE-Top-To-Input-SAGE | **86.46**±0.4 | 85.51±1.1 | 84.05±0.7 | 84.71±0.6 |
| Backpropagation-GAT | 81.36±0.6 | 78.85±0.4 | 78.59±0.4 | 78.31±0.4 |
| CaFo-GAT | 77.25±0.7 | 81.92±3.1 | 81.78±3.2 | 82.06±3.0 |
| ForwardGNN-SymBa-GAT | 91.96±0.3 | 92.01±0.3 | 91.97±0.3 | 92.01±0.3 |
| ForwardGNN-FF-GAT | 92.33±0.3 | **92.36**±0.3 | **92.36**±0.3 | **92.36**±0.3 |
| ForwardGNN-CE-GAT | 89.22±0.3 | 90.36±0.2 | 90.53±0.2 | 90.36±0.2 |
| ForwardGNN-CE-Top-To-Input-GAT | **92.38**±0.2 | 89.14±0.2 | 89.71±0.2 | 89.30±0.4 |

Table 5: *(Continued from the previous table)* Link prediction performance of GCN, SAGE, and GAT averaged over five runs on five real-world graphs, obtained with a different number of layers, and using different learning methods. The best results are in **bold** font, and the best results among the forward learning methods are in red.

(c) AMAZON

| Method | ROC AUC (↑) | | | |
|---|---|---|---|---|
| | # Layers=1 | # Layers=2 | # Layers=3 | # Layers=4 |
| Backpropagation-GCN | 82.41±0.3 | 81.38±0.3 | 88.69±0.1 | 89.47±0.2 |
| CaFo-GCN | 81.11±1.1 | 94.02±0.6 | 94.05±0.6 | 94.00±0.6 |
| ForwardGNN-SymBa-GCN | 82.26±7.4 | 83.95±6.2 | 84.06±6.4 | 83.95±6.2 |
| ForwardGNN-FF-GCN | 96.18±0.2 | 96.16±0.2 | 96.17±0.2 | **96.16**±0.2 |
| ForwardGNN-CE-GCN | 93.84±0.2 | 95.49±0.2 | 95.38±0.2 | 95.49±0.2 |
| ForwardGNN-CE-Top-To-Input-GCN | **97.13**±0.1 | **96.81**±0.3 | **96.32**±0.6 | 95.84±0.7 |
| Backpropagation-SAGE | 70.73±0.4 | 83.43±0.1 | 86.10±0.2 | 87.07±2.5 |
| CaFo-SAGE | 70.93±1.0 | 75.30±1.9 | 77.28±1.6 | 78.20±1.7 |
| ForwardGNN-SymBa-SAGE | 96.25±0.1 | **96.46**±0.1 | **96.46**±0.1 | **96.46**±0.1 |
| ForwardGNN-FF-SAGE | 94.07±0.5 | 94.94±0.6 | 94.86±0.5 | 94.94±0.6 |
| ForwardGNN-CE-SAGE | 93.28±0.1 | 94.96±0.1 | 94.88±0.1 | 94.96±0.1 |
| ForwardGNN-CE-Top-To-Input-SAGE | **97.13**±0.1 | 95.11±0.6 | 95.15±0.6 | 95.12±0.6 |
| Backpropagation-GAT | 71.67±0.8 | 68.54±1.2 | 67.02±1.8 | 66.50±1.8 |
| CaFo-GAT | 73.73±1.1 | 88.09±3.9 | 90.58±2.1 | 91.38±2.3 |
| ForwardGNN-SymBa-GAT | **97.09**±0.1 | 94.86±4.4 | **97.01**±0.1 | **94.86**±4.4 |
| ForwardGNN-FF-GAT | 94.00±1.2 | 94.10±1.1 | 94.10±1.1 | 94.10±1.1 |
| ForwardGNN-CE-GAT | 92.31±0.5 | 94.34±0.2 | 94.36±0.2 | 94.34±0.2 |
| ForwardGNN-CE-Top-To-Input-GAT | 96.90±0.1 | **96.97**±0.6 | 93.02±0.9 | 93.41±0.6 |

(d) CORAML

| Method | ROC AUC (↑) | | | |
|---|---|---|---|---|
| | # Layers=1 | # Layers=2 | # Layers=3 | # Layers=4 |
| Backpropagation-GCN | 82.20±0.3 | 79.86±0.7 | 79.97±0.6 | 79.71±0.5 |
| CaFo-GCN | 73.23±0.6 | 83.88±3.2 | 85.26±2.7 | 85.41±2.8 |
| ForwardGNN-SymBa-GCN | 87.68±5.3 | 87.52±5.6 | 88.02±5.8 | 87.52±5.6 |
| ForwardGNN-FF-GCN | 91.91±1.3 | 92.46±0.9 | 92.39±1.1 | 92.46±0.9 |
| ForwardGNN-CE-GCN | 91.77±0.3 | **93.30**±0.3 | **93.26**±0.3 | **93.30**±0.3 |
| ForwardGNN-CE-Top-To-Input-GCN | **93.64**±0.2 | 91.03±0.6 | 91.57±0.4 | 91.53±0.9 |
| Backpropagation-SAGE | 65.48±0.6 | 76.75±2.0 | 74.65±0.5 | 74.21±0.8 |
| CaFo-SAGE | 63.28±0.9 | 67.46±2.3 | 69.46±0.7 | 70.67±0.5 |
| ForwardGNN-SymBa-SAGE | 71.12±8.7 | 78.22±1.8 | 78.00±1.9 | 78.22±1.8 |
| ForwardGNN-FF-SAGE | 89.06±0.5 | 88.96±0.5 | 89.00±0.5 | 88.96±0.5 |
| ForwardGNN-CE-SAGE | **90.36**±0.6 | **91.96**±0.5 | **91.93**±0.4 | **91.96**±0.5 |
| ForwardGNN-CE-Top-To-Input-SAGE | 89.39±0.3 | 89.72±0.5 | 89.32±0.5 | 89.18±0.4 |
| Backpropagation-GAT | 76.75±0.8 | 77.69±1.0 | 69.61±0.3 | 66.04±0.7 |
| CaFo-GAT | 71.80±0.6 | 83.29±2.3 | 84.71±1.3 | 85.06±1.1 |
| ForwardGNN-SymBa-GAT | 80.41±10.8 | 85.03±3.6 | 85.04±3.6 | 85.03±3.6 |
| ForwardGNN-FF-GAT | 89.42±0.8 | 90.45±0.9 | 90.45±0.9 | 90.45±0.9 |
| ForwardGNN-CE-GAT | 88.96±0.4 | **90.98**±0.4 | **90.96**±0.5 | **90.98**±0.4 |
| ForwardGNN-CE-Top-To-Input-GAT | **91.07**±0.4 | 88.87±0.8 | 89.90±0.5 | 89.70±0.6 |

Table 5: *(Continued from the previous table)* Link prediction performance of GCN, SAGE, and GAT averaged over five runs on five real-world graphs, obtained with a different number of layers, and using different learning methods. The best results are in **bold** font, and the best results among the forward learning methods are in red.

(e) GITHUB

| Method | ROC AUC (↑) | | | |
|---|---|---|---|---|
| | # Layers=1 | # Layers=2 | # Layers=3 | # Layers=4 |
| Backpropagation-GCN | 89.60±0.1 | 91.36±0.1 | 91.71±0.1 | 91.79±0.1 |
| CaFo-GCN | 91.10±0.1 | 91.42±0.1 | 91.32±0.2 | 91.24±0.2 |
| ForwardGNN-SymBa-GCN | 81.41±15.7 | 81.51±15.8 | 81.51±15.8 | 81.51±15.8 |
| ForwardGNN-FF-GCN | 93.27±0.1 | **93.48**±0.0 | **93.48**±0.0 | **93.48**±0.0 |
| ForwardGNN-CE-GCN | 91.87±0.1 | 91.30±0.1 | 91.45±0.1 | 91.30±0.1 |
| ForwardGNN-CE-Top-To-Input-GCN | **94.09**±0.1 | 91.82±0.4 | 91.44±0.4 | 91.64±0.5 |
| Backpropagation-SAGE | 83.85±0.1 | 84.45±0.2 | 85.03±0.8 | 86.90±0.2 |
| CaFo-SAGE | 76.63±0.9 | 77.21±1.0 | 77.29±0.9 | 77.37±0.9 |
| ForwardGNN-SymBa-SAGE | **93.88**±0.1 | **92.55**±2.3 | **93.69**±0.0 | **92.55**±2.3 |
| ForwardGNN-FF-SAGE | 89.64±0.3 | 89.62±0.3 | 89.63±0.3 | 89.62±0.3 |
| ForwardGNN-CE-SAGE | 87.17±0.3 | 88.35±0.3 | 88.29±0.4 | 88.35±0.3 |
| ForwardGNN-CE-Top-To-Input-SAGE | 88.93±0.6 | 84.45±1.3 | 84.43±0.9 | 83.88±0.7 |
| Backpropagation-GAT | 78.75±0.4 | 73.41±1.1 | 73.94±1.4 | 68.26±0.5 |
| CaFo-GAT | 81.88±0.9 | 82.66±0.6 | 83.09±0.7 | 83.12±0.7 |
| ForwardGNN-SymBa-GAT | **92.99**±0.2 | **92.53**±0.2 | **92.49**±0.2 | **92.53**±0.2 |
| ForwardGNN-FF-GAT | 88.18±0.6 | 88.63±0.8 | 88.63±0.8 | 88.63±0.8 |
| ForwardGNN-CE-GAT | 82.72±0.4 | 82.10±0.5 | 82.13±0.6 | 82.10±0.5 |
| ForwardGNN-CE-Top-To-Input-GAT | 89.59±0.2 | 80.91±1.2 | 81.39±0.3 | 81.81±0.4 |

Table 6: GPU memory usage (in MB) for link prediction. The best results are in **bold** font, and the best results among the forward learning methods are in red.

(a) CITESEER

| Method | GPU Memory Usage (↓) | | | |
|---|---|---|---|---|
| | # Layers=1 | # Layers=2 | # Layers=3 | # Layers=4 |
| Backpropagation-GCN | **10.16** | 13.48 | 16.41 | 18.14 |
| CaFo-GCN | 13.19 | 13.19 | 13.24 | 13.24 |
| ForwardGNN-SymBa-GCN | 17.82 | 18.04 | 18.04 | 18.04 |
| ForwardGNN-FF-GCN | 17.84 | 18.06 | 18.06 | 18.06 |
| ForwardGNN-CE-GCN | 14.47 | 14.47 | 14.47 | 14.47 |
| ForwardGNN-CE-Top-To-Input-GCN | 21.64 | 24.79 | 24.61 | 25.01 |
| Backpropagation-SAGE | 22.38 | 26.28 | 30.13 | 34.78 |
| CaFo-SAGE | 13.44 | 13.44 | 13.62 | 13.62 |
| ForwardGNN-SymBa-SAGE | 29.07 | 29.07 | 29.07 | 29.07 |
| ForwardGNN-FF-SAGE | 29.09 | 29.09 | 29.09 | 29.09 |
| ForwardGNN-CE-SAGE | 24.17 | 24.17 | 24.17 | 24.17 |
| ForwardGNN-CE-Top-To-Input-SAGE | 32.93 | 39.41 | 39.78 | 39.03 |
| Backpropagation-GAT | 18.39 | 30.70 | 41.60 | 53.16 |
| CaFo-GAT | 13.20 | 13.20 | 13.20 | 13.20 |
| ForwardGNN-SymBa-GAT | 25.66 | 26.23 | 26.23 | 26.23 |
| ForwardGNN-FF-GAT | 25.68 | 26.25 | 26.25 | 26.25 |
| ForwardGNN-CE-GAT | 21.91 | 21.91 | 21.91 | 21.91 |
| ForwardGNN-CE-Top-To-Input-GAT | 31.61 | 33.78 | 33.03 | 32.89 |

Table 6: *(Continued from the previous table)* GPU memory usage (in MB) for link prediction. The best results are in **bold** font, and the best results among the forward learning methods are in red.

(b) PUBMED

| Method | GPU Memory Usage (↓) | | | |
|---|---|---|---|---|
| | # Layers=1 | # Layers=2 | # Layers=3 | # Layers=4 |
| Backpropagation-GCN | **68.44** | 79.70 | 90.37 | 103.04 |
| CaFo-GCN | 76.86 | **76.86** | **76.86** | **76.86** |
| ForwardGNN-SymBa-GCN | 123.38 | 123.38 | 123.38 | 123.38 |
| ForwardGNN-FF-GCN | 123.59 | 123.59 | 123.59 | 123.59 |
| ForwardGNN-CE-GCN | 95.39 | 95.39 | 95.39 | 95.39 |
| ForwardGNN-CE-Top-To-Input-GCN | 106.89 | 118.13 | 118.13 | 118.05 |
| Backpropagation-SAGE | 104.82 | 124.65 | 144.48 | 164.31 |
| CaFo-SAGE | **77.76** | **77.76** | **77.76** | **77.76** |
| ForwardGNN-SymBa-SAGE | 161.00 | 161.00 | 161.00 | 161.00 |
| ForwardGNN-FF-SAGE | 161.22 | 161.22 | 161.22 | 161.22 |
| ForwardGNN-CE-SAGE | 133.19 | 133.19 | 133.19 | 133.19 |
| ForwardGNN-CE-Top-To-Input-SAGE | 144.99 | 166.28 | 166.36 | 165.61 |
| Backpropagation-GAT | 121.26 | 184.54 | 249.24 | 313.18 |
| CaFo-GAT | **76.11** | **76.11** | **76.86** | **76.86** |
| ForwardGNN-SymBa-GAT | 176.70 | 176.70 | 176.70 | 176.70 |
| ForwardGNN-FF-GAT | 176.92 | 176.92 | 176.92 | 176.92 |
| ForwardGNN-CE-GAT | 148.60 | 148.60 | 148.60 | 148.60 |
| ForwardGNN-CE-Top-To-Input-GAT | 160.54 | 170.59 | 171.04 | 170.29 |

(c) AMAZON

| Method | GPU Memory Usage (↓) | | | |
|---|---|---|---|---|
| | # Layers=1 | # Layers=2 | # Layers=3 | # Layers=4 |
| Backpropagation-GCN | **157.68** | 164.72 | 171.76 | 178.80 |
| CaFo-GCN | 158.62 | **158.62** | **158.62** | **158.62** |
| ForwardGNN-SymBa-GCN | 307.27 | 307.27 | 307.27 | 307.27 |
| ForwardGNN-FF-GCN | 307.85 | 307.85 | 307.85 | 307.85 |
| ForwardGNN-CE-GCN | 232.11 | 232.55 | 232.55 | 232.55 |
| ForwardGNN-CE-Top-To-Input-GCN | 182.17 | 186.42 | 186.42 | 186.47 |
| Backpropagation-SAGE | 177.82 | 185.83 | 193.83 | 201.83 |
| CaFo-SAGE | **160.07** | **160.07** | **160.07** | **160.07** |
| ForwardGNN-SymBa-SAGE | 326.97 | 326.97 | 326.97 | 326.97 |
| ForwardGNN-FF-SAGE | 327.55 | 327.55 | 327.55 | 327.55 |
| ForwardGNN-CE-SAGE | 252.25 | 252.25 | 252.25 | 252.25 |
| ForwardGNN-CE-Top-To-Input-SAGE | 203.32 | 212.05 | 212.05 | 212.05 |
| Backpropagation-GAT | 249.88 | 348.22 | 446.55 | 544.89 |
| CaFo-GAT | **158.62** | **158.62** | **158.62** | **158.79** |
| ForwardGNN-SymBa-GAT | 400.60 | 400.60 | 400.60 | 400.60 |
| ForwardGNN-FF-GAT | 401.18 | 401.18 | 401.18 | 401.18 |
| ForwardGNN-CE-GAT | 324.31 | 324.31 | 324.31 | 324.31 |
| ForwardGNN-CE-Top-To-Input-GAT | 273.61 | 277.97 | 277.97 | 277.86 |

Table 6: *(Continued from the previous table)* GPU memory usage (in MB) for link prediction. The best results are in **bold** font, and the best results among the forward learning methods are in red.

(d) CoraML

| Method | GPU Memory Usage (↓) | | | |
|---|---|---|---|---|
| | # Layers=1 | # Layers=2 | # Layers=3 | # Layers=4 |
| Backpropagation-GCN | **17.58** | **19.55** | 21.65 | 23.62 |
| CaFo-GCN | 19.92 | 19.92 | **19.92** | **19.92** |
| ForwardGNN-SymBa-GCN | 28.42 | 28.42 | 28.42 | 28.42 |
| ForwardGNN-FF-GCN | 28.46 | 28.46 | 28.46 | 28.46 |
| ForwardGNN-CE-GCN | 22.68 | 22.68 | 22.68 | 22.68 |
| ForwardGNN-CE-Top-To-Input-GCN | 61.34 | 61.34 | 61.36 | 62.08 |
| Backpropagation-SAGE | 55.84 | 59.28 | 63.43 | 66.87 |
| CaFo-SAGE | **24.66** | **25.00** | **25.00** | **25.00** |
| ForwardGNN-SymBa-SAGE | 66.06 | 66.06 | 66.06 | 66.06 |
| ForwardGNN-FF-SAGE | 66.10 | 66.10 | 66.10 | 66.10 |
| ForwardGNN-CE-SAGE | 60.94 | 60.94 | 60.94 | 60.94 |
| ForwardGNN-CE-Top-To-Input-SAGE | 100.92 | 107.69 | 107.72 | 104.29 |
| Backpropagation-GAT | 26.47 | 37.33 | 48.19 | 59.05 |
| CaFo-GAT | **19.05** | **19.05** | **19.05** | **19.05** |
| ForwardGNN-SymBa-GAT | 37.14 | 37.14 | 37.68 | 38.09 |
| ForwardGNN-FF-GAT | 37.18 | 37.18 | 37.72 | 38.13 |
| ForwardGNN-CE-GAT | 32.18 | 32.18 | 32.18 | 32.18 |
| ForwardGNN-CE-Top-To-Input-GAT | 65.54 | 68.17 | 68.17 | 67.25 |

(e) GitHub

| Method | GPU Memory Usage (↓) | | | |
|---|---|---|---|---|
| | # Layers=1 | # Layers=2 | # Layers=3 | # Layers=4 |
| Backpropagation-GCN | **390.25** | 416.72 | 442.69 | 469.09 |
| CaFo-GCN | 400.17 | **400.17** | **400.17** | **400.17** |
| ForwardGNN-SymBa-GCN | 751.07 | 751.07 | 751.07 | 751.07 |
| ForwardGNN-FF-GCN | 752.48 | 752.48 | 752.48 | 752.48 |
| ForwardGNN-CE-GCN | 569.88 | 569.88 | 569.88 | 569.88 |
| ForwardGNN-CE-Top-To-Input-GCN | 409.54 | 428.44 | 428.48 | 428.48 |
| Backpropagation-SAGE | **400.17** | 437.63 | 475.09 | 512.55 |
| CaFo-SAGE | 401.16 | **401.16** | **401.16** | **401.16** |
| ForwardGNN-SymBa-SAGE | 762.69 | 762.69 | 762.69 | 762.69 |
| ForwardGNN-FF-SAGE | 764.10 | 764.10 | 764.10 | 764.10 |
| ForwardGNN-CE-SAGE | 581.54 | 581.54 | 581.54 | 581.54 |
| ForwardGNN-CE-Top-To-Input-SAGE | 420.49 | 459.05 | 459.05 | 459.23 |
| Backpropagation-GAT | 633.29 | 901.83 | 1171.35 | 1441.82 |
| CaFo-GAT | **400.17** | **400.17** | **400.17** | **400.17** |
| ForwardGNN-SymBa-GAT | 994.10 | 994.10 | 994.10 | 994.10 |
| ForwardGNN-FF-GAT | 995.52 | 995.52 | 995.52 | 995.52 |
| ForwardGNN-CE-GAT | 812.95 | 812.95 | 812.95 | 812.95 |
| ForwardGNN-CE-Top-To-Input-GAT | 652.58 | 671.37 | 671.37 | 671.37 |

## F.3 TRAINING TIME

To evaluate the training efficiency of different learning approaches, we report in Table 7 the time taken to train GNNs for 100 epochs for node classification on GITHUB and CITESEER, with a varying number of GNN layers. From Table 7, we make the following observations.

- **The proposed forward learning approaches in Sections 3.1 and 3.2 train GCN and SAGE models much faster than backpropagation (up to 9 times when using four layers)**. With these greedy forward learning approaches, such as those presented in Sections 3.1 and 3.2, GNN training can be made much more efficient than with backpropagation by exploiting the fact that the neighborhood aggregation procedure can be performed just once while the parameters of each GNN layer are optimized over multiple training epochs. This is because, similar to You et al. (2020), the input to each layer (*i.e.*, the output from the lower layer) remains the same throughout the training of the layer, and thus the neighborhood aggregation results also do not change due to the way the greedy forward training scheme and GNNs work. This way, only the feature update step after the aggregation needs to be performed multiple times during the training, while the aggregation is done only once, which can lead to a significant speedup in training GNNs, especially for large-scale graphs.

- **While the training with forward learning approaches takes longer than BP in some cases, the increase in training time in such cases is still modest in comparison to BP (*e.g.*, up to 1.6 times longer time to train four-layer GCNs on GITHUB)**. With forward learning approaches with top-down signal paths, or with models like GAT, the aforementioned speedup cannot be used, as the output from the neighborhood aggregation procedure does no longer remain the same throughout the training of the layer, due to the changes in the input resulting from the incorporation of top-down signals, or due to the changes in the parameters used for attentive neighborhood aggregation. Still, the training time increase in such cases is not that significant, and the forward learning methods can scale up in a much more memory-efficient manner than BP.

- **The single-forward (SF) approach improves the training efficiency of the forward-forward based approaches** as SF avoids the computational overhead to perform multiple forward passes.

Table 7: Training time (in seconds) for node classification. The best results are in **bold** font, and the best results among the forward learning methods are in red.

### (a) GITHUB

| Method | Training time (seconds) (↓) | | | |
|---|---|---|---|---|
| | # Layers=1 | # Layers=2 | # Layers=3 | # Layers=4 |
| Backpropagation-GCN | 1.77±1.7 | 5.54±1.7 | 10.37±3.2 | 13.04±1.7 |
| SymBa-GCN | 1.41±1.0 | 2.09±1.0 | 2.78±1.0 | 3.48±1.0 |
| PEPITA-GCN | 1.40±1.0 | 7.15±0.8 | 13.09±0.8 | 18.95±0.8 |
| FF-LA-GCN (Sec 3.1) | 1.47±1.2 | 2.06±1.2 | 2.65±1.3 | 3.24±1.3 |
| FF-VN-GCN (Sec 3.1) | 1.47±1.0 | 2.19±1.0 | 2.91±1.0 | 3.64±1.1 |
| SF-GCN (Sec 3.2) | **1.10**±0.9 | **1.57**±0.9 | **2.04**±0.9 | **2.52**±0.9 |
| SF-Top-To-Loss-GCN (Sec 3.3) | 5.66±0.8 | 11.23±1.5 | 16.06±1.1 | 21.15±1.1 |
| SF-Top-To-Input-GCN (Sec 3.3) | 5.67±0.8 | 10.80±0.9 | 15.91±0.8 | 21.12±1.1 |
| Backpropagation-SAGE | 4.54±2.4 | 7.99±2.3 | 11.01±1.6 | 14.91±2.1 |
| SymBa-SAGE | 1.28±1.4 | 1.55±1.4 | 1.82±1.4 | 2.09±1.4 |
| PEPITA-SAGE | 6.62±1.4 | 11.74±0.9 | 17.58±1.5 | 22.50±0.7 |
| FF-LA-SAGE (Sec 3.1) | 1.31±1.3 | 1.68±1.4 | 2.04±1.4 | 2.40±1.3 |
| FF-VN-SAGE (Sec 3.1) | 0.83±0.7 | 1.11±0.7 | 1.38±0.7 | 1.66±0.7 |
| SF-SAGE (Sec 3.2) | **0.82**±0.7 | **1.08**±0.7 | **1.34**±0.7 | **1.61**±0.7 |
| SF-Top-To-Loss-SAGE (Sec 3.3) | 4.60±1.5 | 7.53±0.8 | 10.99±0.8 | 14.61±1.0 |
| SF-Top-To-Input-SAGE (Sec 3.3) | 4.08±0.8 | 10.44±0.8 | 16.76±0.8 | 23.11±0.8 |
| Backpropagation-GAT | 1.89±2.4 | 1.79±1.7 | 3.45±3.4 | 2.64±1.5 |
| SymBa-GAT | 1.72±0.8 | 2.85±0.8 | 3.98±0.8 | 5.12±0.8 |
| PEPITA-GAT | **0.83**±0.8 | **1.13**±0.8 | **1.63**±0.9 | **1.85**±0.8 |
| FF-LA-GAT (Sec 3.1) | 1.70±0.9 | 2.78±0.9 | 3.85±0.9 | 4.93±0.9 |
| FF-VN-GAT (Sec 3.1) | 1.75±0.8 | 2.90±0.8 | 4.06±0.8 | 5.24±0.8 |
| SF-GAT (Sec 3.2) | 1.21±0.8 | 1.89±0.8 | 2.56±0.8 | 3.24±0.8 |
| SF-Top-To-Loss-GAT (Sec 3.3) | 1.59±1.3 | 2.36±1.4 | 2.70±1.0 | 3.35±0.9 |
| SF-Top-To-Input-GAT (Sec 3.3) | 1.25±0.8 | 2.41±1.5 | 3.45±2.0 | 3.24±0.8 |

### (b) CITESEER

| Method | Training time (seconds) (↓) | | | |
|---|---|---|---|---|
| | # Layers=1 | # Layers=2 | # Layers=3 | # Layers=4 |
| Backpropagation-GCN | 2.17±2.5 | 2.33±2.8 | 1.81±1.8 | **1.88**±1.7 |
| SymBa-GCN | 1.96±1.5 | 2.81±1.5 | 3.67±1.5 | 4.54±1.5 |
| PEPITA-GCN | 1.02±1.0 | 1.33±1.0 | 1.84±1.3 | 2.16±1.5 |
| FF-LA-GCN (Sec 3.1) | 1.91±1.5 | 2.71±1.5 | 3.52±1.5 | 4.34±1.5 |
| FF-VN-GCN (Sec 3.1) | 2.09±1.7 | 3.00±1.7 | 3.92±1.7 | 4.83±1.8 |
| SF-GCN (Sec 3.2) | 1.44±1.7 | 1.67±1.7 | 1.90±1.7 | 2.13±1.7 |
| SF-Top-To-Loss-GCN (Sec 3.3) | 0.93±0.8 | 1.27±0.7 | 1.68±0.8 | 1.97±0.8 |
| SF-Top-To-Input-GCN (Sec 3.3) | 1.08±1.0 | 1.55±1.2 | 2.14±1.5 | 2.47±1.5 |
| Backpropagation-SAGE | 2.03±2.3 | 2.51±2.7 | 2.62±2.7 | 2.24±2.0 |
| SymBa-SAGE | 2.21±2.1 | 2.90±2.1 | 3.59±2.1 | 4.27±2.1 |
| PEPITA-SAGE | 1.27±0.8 | 1.49±0.9 | 1.66±0.9 | 2.47±1.7 |
| FF-LA-SAGE (Sec 3.1) | 1.90±1.7 | 2.57±1.7 | 3.19±1.8 | 3.83±1.9 |
| FF-VN-SAGE (Sec 3.1) | 1.34±1.0 | 1.92±1.0 | 2.49±1.0 | 3.07±1.0 |
| SF-SAGE (Sec 3.2) | 1.24±1.4 | 1.43±1.4 | 1.62±1.4 | 1.81±1.4 |
| SF-Top-To-Loss-SAGE (Sec 3.3) | 1.15±0.7 | 2.03±1.5 | 1.78±0.8 | 2.04±0.7 |
| SF-Top-To-Input-SAGE (Sec 3.3) | 1.22±0.8 | 2.44±2.0 | 1.93±0.8 | 2.28±0.7 |
| Backpropagation-GAT | 1.41±1.8 | 2.30±2.9 | 1.69±1.9 | 1.64±1.7 |
| SymBa-GAT | 1.52±0.7 | 2.43±0.7 | 3.31±0.7 | 4.21±0.7 |
| PEPITA-GAT | **0.76**±0.8 | **0.89**±0.8 | 1.50±1.4 | **1.16**±0.8 |
| FF-LA-GAT (Sec 3.1) | 1.71±1.0 | 2.67±1.0 | 3.62±1.1 | 4.61±1.1 |
| FF-VN-GAT (Sec 3.1) | 1.46±0.8 | 2.29±0.9 | 3.13±0.9 | 3.97±0.9 |
| SF-GAT (Sec 3.2) | 1.09±1.2 | 1.32±1.2 | 1.55±1.2 | 1.77±1.2 |
| SF-Top-To-Loss-GAT (Sec 3.3) | 0.82±0.8 | 1.05±0.8 | 1.75±1.4 | 1.55±0.8 |
| SF-Top-To-Input-GAT (Sec 3.3) | 0.81±0.7 | 1.03±0.8 | **1.20**±0.7 | 1.41±0.7 |

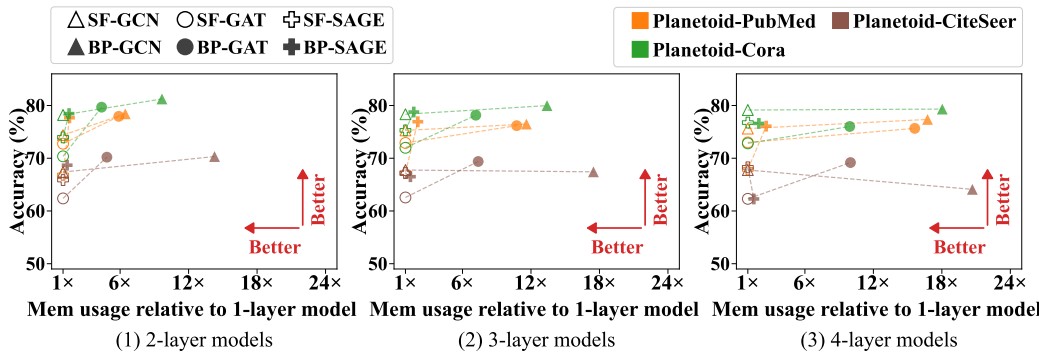

(1) 2-layer models    (2) 3-layer models    (3) 4-layer models

Figure 7: The single-forward approach (SF, empty symbols) vs. backpropagation (BP, filled symbols) on node classification in terms of classification accuracy (y-axis) vs. memory usage increase (x-axis). Lines towards the lower right corner (SF outperforms BP), and upper right corner (BP outperforms SF); horizontal lines (both perform the same).

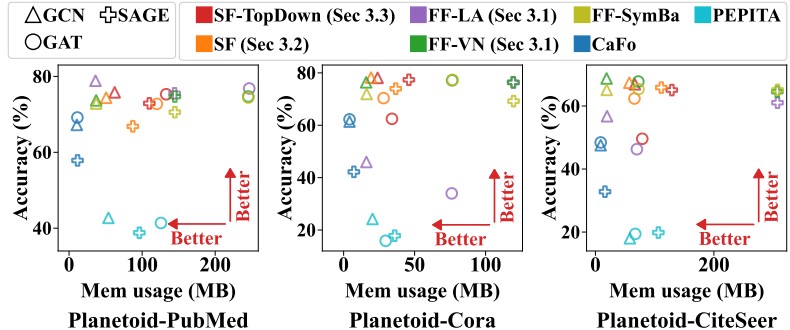

Planetoid-PubMed    Planetoid-Cora    Planetoid-CiteSeer

Figure 8: Node classification accuracy (y-axis) and memory usage (x-axis) on Planetoid datasets (Table 8) as three GNNs (denoted by symbol shape) are trained with different forward learning approaches (denoted by symbol color).

Table 8: Summary statistics of the graph datasets with limited training data.

| Dataset | # Nodes | # Edges | # Training Nodes | # Validation Nodes | # Testing Nodes | # Features | # Classes |
|---|---|---|---|---|---|---|---|
| PLANETOID-CORA | 2,708 | 10,556 | 140 | 500 | 1,000 | 1,433 | 7 |
| PLANETOID-CITESEER | 3,327 | 9,104 | 120 | 500 | 1,000 | 3,703 | 6 |
| PLANETOID-PUBMED | 19,717 | 88,648 | 60 | 500 | 1,000 | 500 | 3 |

## F.4   NODE CLASSIFICATION RESULTS WITH LIMITED TRAINING DATA

In this section, we evaluate how well the proposed forward learning approach performs in comparison to BP when given a small amount of training data. To that end, we use three citation networks, namely, PLANETOID-CORA, PLANETOID-PUBMED, and PLANETOID-CITESEER listed in Table 8, which provide much less amount of training data than the graphs used in the main text (Table 2).

Figure 7 shows the classification accuracy (y-axis) versus memory usage increase (x-axis) obtained with the single forward (SF) and backpropagation (BP) algorithms on the three graphs with limited training data. From Figure 7, we make the following observations.

- In these datasets with sparse training data, the proposed SF (empty symbols) perform similarly to BP (filled symbols), or even outperforms BP in a few cases, *e.g.*, when four GNN layers are used. These results are similar to the results obtained in the setup with richer training data (Figure 2a).

- With two layers (Figure 7(1)), BP mostly outperforms SF, which is different from the results in the counterpart in Figure 2a(1), where BP and SF performed similarly. The contrastive single forward learning mechanism tends to require more training data than BP to arrive at a similar level of predictive accuracy, but the gap closes as more layers are used.

- SF shows a near-constant memory usage, while the relative memory usage of BP keeps increasing (up to around $21\times$). as the number of layers increases.

Figure 8 shows the node classification accuracy (y-axis) versus memory usage (x-axis), as GNNs (denoted by the symbol shape) are trained with different forward learning approaches (denoted by the symbol color). Overall, Figure 8 shows a similar pattern to Figure 3, which shows the performance of forward learning approaches on datasets with richer training labels. Thus our discussion on Figure 3 in Section 4.3 similarly applies to Figure 8.

### F.5 IMPACT OF THE DIRECTIONALITY OF EDGES BETWEEN REAL AND VIRTUAL NODES

In the main text, the results of the proposed approaches that employ virtual nodes were obtained using the augmented graph where real nodes and virtual nodes are connected via bidirectional edges (*i.e.*, in both ways between the real and virtual nodes). Since virtual nodes are processed in the same way as the real nodes, they aggregate information from a large number of real nodes, and then propagate aggregated information back to those real nodes, which can potentially adversely impact the classification performance. Here we evaluate a different approach to utilize virtual nodes, which is to connect the real and virtual nodes via unidirectional edges (*i.e.*, in one way from the real to the virtual nodes), such that virtual nodes only receive information from the real nodes, without sending the aggregated message back to them. In Figure 9, we report the node classification performance when the edges between the real and virtual nodes are bidirectional or unidirectional, obtained with (a) the forward-forward (FF) approach (Section 3.1), (b) the single forward (SF) approach (Section 3.2), and (c-d) the SF approaches with the top-down signal path (Section 3.3). Results show that the impact of using unidirectional edges, in comparison to when bidirectional edges are used, varies a lot among forward learning approaches. With the FF approach (Figure 9(a)) and the SF approach with the top-to-loss signal path (Figure 9(d)), using bidirectional edges led to better results than using unidirectional edges in most cases. In particular, for the FF approach, the information from the virtual nodes to the real nodes, which lacks when unidirectional edges are used, tends to play an important role for the model to learn to distinguish between correct and incorrect classes. By contrast, with the SF approach (Figure 9(b)), using unidirectional edges led to better performance in several cases, especially on PUBMED and GITHUB. With the SF approach that employs the top-to-input signal path (Figure 9(c)), both types of edges performed similarly to each other.

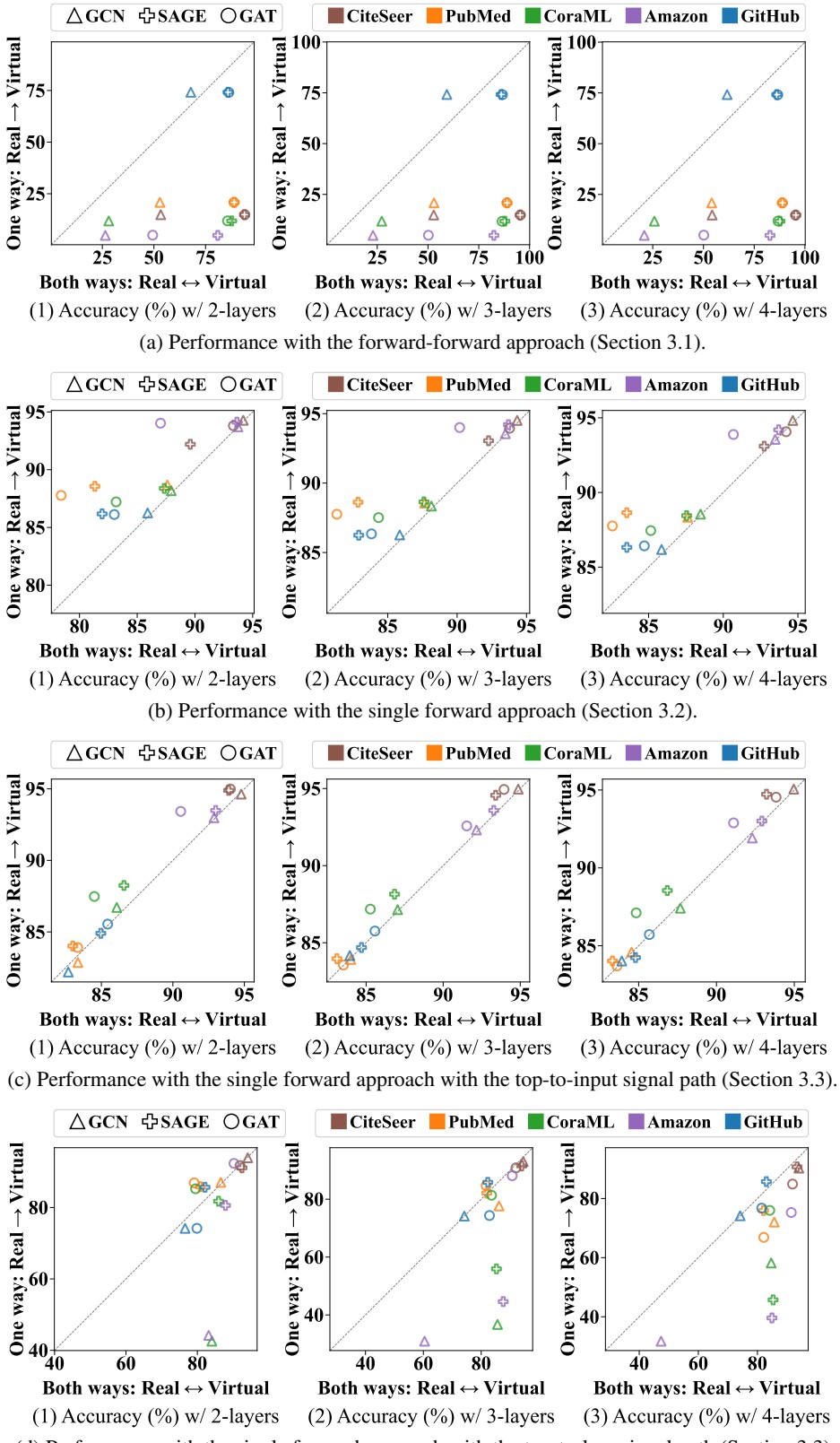

(1) Accuracy (%) w/ 2-layers    (2) Accuracy (%) w/ 3-layers    (3) Accuracy (%) w/ 4-layers

(a) Performance with the forward-forward approach (Section 3.1).

(1) Accuracy (%) w/ 2-layers    (2) Accuracy (%) w/ 3-layers    (3) Accuracy (%) w/ 4-layers

(b) Performance with the single forward approach (Section 3.2).

(1) Accuracy (%) w/ 2-layers    (2) Accuracy (%) w/ 3-layers    (3) Accuracy (%) w/ 4-layers

(c) Performance with the single forward approach with the top-to-input signal path (Section 3.3).

(1) Accuracy (%) w/ 2-layers    (2) Accuracy (%) w/ 3-layers    (3) Accuracy (%) w/ 4-layers

(d) Performance with the single forward approach with the top-to-loss signal path (Section 3.3).

Figure 9: Node classification accuracy using virtual node-based forward learning approaches when the edges between the real and virtual nodes are bidirectional (*i.e.*, in both ways between the real and virtual nodes), and unidirectional (*i.e.*, in one way from the real to the virtual nodes), obtained with (a) the forward-forward (FF) approach (Section 3.1), (b) the single forward (SF) approach (Section 3.2), and (c-d) the SF approaches with the top-down signal path (Section 3.3).

