# OpenReview forum: "Forward Learning of Graph Neural Networks"
_ICLR.cc/2024/Conference — ICLR 2024 poster_

### Official Review · Reviewer_TBYv · 2023-10-16

**Soundness:** 4 excellent
**Presentation:** 2 fair
**Contribution:** 3 good
**Rating:** 6
**Confidence:** 3

**Summary:**

The paper proposes ForwardGNN, a novel forward learning framework for Graph Neural Networks (GNNs) that addresses the limitations of the backpropagation (BP) algorithm, such as memory overhead and biological implausibility. By building upon and improving the forward-forward algorithm (FF), ForwardGNN is designed to work with graph data and GNNs without generating negative inputs. This results in a more efficient training and inference process. The framework also enables each layer to learn from both bottom-up and top-down signals without relying on backpropagation. Experiments conducted on five real-world datasets and three representative GNNs demonstrate the effectiveness and generality of the forward graph learning framework, showing that it outperforms or performs on par with BP while using less memory for training.

**Strengths:**

1. This paper systematically explores the potential of forward graph learning, paving the way for biologically plausible optimization techniques in GNNs.

2. The performance is impressive; this marks the first instance where FF algorithms outperform BP-trained deep neural networks in real-world applications.

3. The study proposes numerous algorithms that are model-agnostic, potentially inspiring further research on forward-forward algorithms in various applications.

**Weaknesses:**

There are a few areas for improvement:

1. The proposed methods share a close relationship with layer-wise training of neural networks, which could potentially diminish the significance of this paper. (Refer to Question 1)

2. The data splitting does not adhere to standard practices. (Please see Question 2)

3. The presentation is difficult to follow and should be improved. (Please refer to Question 3,4,5)

4. Table 2, and 3 contains many interesting results but lacks explanations and intuitions. (Please refer to Questions 6, 7, 8)

**Questions:**

1. The proposed method involves layer-by-layer training of GNNs. In the literature, layer-wise training of deep neural networks has been a long-standing topic [1,2]. A more extensive literature review is necessary to differentiate the present paper. Additionally, the paper's contribution could be strengthened by including experiments for comparison with [2].

2. The data splitting differs significantly from those in previous papers. I recommend that the authors conduct experiments using standard splitting for semi-supervised node classification tasks.

3. In Figure 2, the upper-left corner is crowded with too many methods, making it difficult to read. Presenting the results in a table might be a better approach (the table should be much simpler compared to Table 2 and Table 3).

4. In Algorithms 1-4, the authors state "optimize layer using the computed loss" but the details of the optimizer are not clear, e.g., what optimizer is used.

5. In addition to memory, training time is an important metric. Including the training time of the proposed method and baselines will complete the picture.

6. In Table 2, why does SF-Top-To-Loss achieve state-of-the-art performance with one layer, but the performance degrades with more layers? An explanation is needed.

7. In Table 2, the best-performing method varies significantly across datasets. Providing an intuition for this variation would be beneficial.

8. In Table 3, the memory of some proposed methods is higher than BP (even 20 times higher in Table 3 (c) GAT). Would this be a significant bottleneck?

[1] Belilovsky E, Eickenberg M, Oyallon E. Greedy layerwise learning can scale to imagenet, ICML 2019.
[2] You Y, Chen T, Wang Z, et al. L2-gcn: Layer-wise and learned efficient training of graph convolutional networks, CVPR 2020.

---

> ### Author Response · Authors · 2023-11-23
> **Response to Reviewer TBYv**
>
> We appreciate your thoughtful comments. Please find our response to your comments below. We have updated the paper where necessary in light of your comment, which is marked in blue in the revised paper.
>
> > The proposed methods share a close relationship with layer-wise training of neural networks, which could potentially diminish the significance of this paper. (Refer to Question 1)
>
> Thank you for the comment. In Appendix D, we added a discussion on alternative learning approaches based on layer-wise model training. Inspired by the forward-forward algorithm (Hinton, 2022), which was proposed as a biologically plausible alternative to the backpropagation (BP), our paper focuses on the forward-only learning of GNNs. The prior layer-wise approaches for GNNs fail to achieve forward-only learning as they rely on BP for optimization. Further, their learning scheme can consider only the bottom-up signals, but not the top-down signals that ForwardGNN incorporate into the forward learning process. Also, we study the forward learning on link prediction, which was not investigated in prior works. We added the following, more detailed discussion in Appendix D.
>
> “Another line of works investigated layer-wise approaches for learning GNNs. A layer-wise learning idea closely related to our work first appeared in the vision domain (Belilovsky et al., 2019), which proposed a greedy layer-wise learning technique for deep convolutional neural networks (CNNs), and obtained promising results on large-scale datasets such as ImageNet. In Belilovsky et al. (2019), a CNN model gets trained progressively and greedily, i.e., layer by layer from the bottom to the top. To enable a layer-wise training, they formulate and solve an auxiliary problem for each layer, where an auxiliary classifier is attached to the layer, forming a block, and the original CNN layer and the auxiliary classifier in the block are jointly optimized in terms of the training accuracy of the auxiliary classifier. You et al. (2020) extended this approach for a layer-wise training of GNNs, in which the CNN layers are replaced with the GNN layers. Furthermore, You et al. (2020) streamlined the graph convolution operation by decoupling the feature aggregation and transformation, and introduced a learnable RNN controller, which automatically decides when to stop the training of each layer via reinforcement learning. A follow-up work presented by Wang et al. (2020) further improved the efficiency of layer-wise GNN training of You et al. (2020) via the proposed parallel training and lazy update scheme. While these methods train GNNs layer by layer, they still rely on BP, since each block is a multi-layer architecture, and as a result, the error derivatives are propagated across the layers in the block for optimization. By contrast, ForwardGNN operates in the forward-only learning regime, which is the focus of this work. Further, their optimization framework allows only bottom-up signal propagation, and does not provide a way to incorporate the top-down signals. Thus, using the above layer-wise methods, the training of each layer cannot be informed by what the upper layers have learned as in ForwardGNN. Also, these studies only investigate the node classification task, and miss out the other fundamental graph learning task of link prediction. In this work, we systematically explore the potential of forward-only learning for both node classification and link prediction tasks.”
>
> In addition to the layer-wise approaches pointed out by this comment, we further discuss different alternatives to BP for learning GNNs, such as a Lagrangian-based optimization method, in Appendix D.

---

> ### Author Response · Authors · 2023-11-23
> **Response to Reviewer TBYv (Cont.)**
>
> > The data splitting does not adhere to standard practices. (Please see Question 2)
>
> We added Appendix E.4 to evaluate how well the proposed forward learning approach performs in comparison to BP when given a small amount of training data. Specifically, we use three citation networks with limited training data, which have been widely used in earlier works, such as the GCN paper (“Semi-Supervised Classification with Graph Convolutional Networks”). Table 7 shows the number of training/validation/testing nodes for each dataset used for evaluation. In Figure 8, we show the classification accuracy versus memory usage increase, obtained with the single forward (SF) and backpropagation (BP) algorithms on three graphs with limited training data.
> - The proposed SF (empty symbols) perform similarly to BP (filled symbols), or outperforms BP in a few cases, when four GNN layers are used, which is similar to the results obtained in the setup with richer training data (Figure 2a).
> - When two layers are used, BP mostly outperform SF, which is different from the results in the counterpart in Figure 2a(1), where BP and SF performed comparably to each other. The contrastive single forward learning mechanism tends to require more training data than BP to arrive at a similar level of predictive accuracy, but the gap closes as more layers are used.
> - These results point to an interesting research direction to further improve the performance of forward-only GNN training in the regime of sparse training data.
>
> In the final version of this paper, we will include more comprehensive results on datasets with limited training data.
>
>
> > Q3. In Figure 2, the upper-left corner is crowded with too many methods, making it difficult to read. Presenting the results in a table might be a better approach (the table should be much simpler compared to Table 2 and Table 3).
>
> Thank you for the suggestion. We tried to add tables in place of Figure 2, but found it hard to display the information of Figure 2 in the same space, due to the large number of data points to be listed in the table, and the space constraint. Instead, we added an instruction in the caption to interpret Figure 2 more easily.
>
> Two major things to interpret Figure 2 are (1) line angle, and (2) line length.
> - Line angle (for prediction performance):
>   - Lines towards the lower right corner: SF outperforms BP.
>   - Lines towards the upper right corner: BP outperforms SF.
>   - Horizontal lines: Both perform the same.
> - Line length (for memory usage): Longer line length indicates a larger increase in memory usage when more layers are used.
>
>
> > Q4. In Algorithms 1-4, the authors state "optimize layer using the computed loss" but the details of the optimizer are not clear, e.g., what optimizer is used.
>
> The proposed forward learning algorithms can work with any optimizer, such as SGD and Adam, among others. To make this point clearer, we revised the corresponding lines in the algorithm listings.

---

> ### Author Response · Authors · 2023-11-23
> **Response to Reviewer TBYv (Cont.)**
>
> > Q5. In addition to memory, training time is an important metric. Including the training time of the proposed method and baselines will complete the picture.
>
> To evaluate the training efficiency of different learning approaches, we added additional results in Table 6 in Appendix E.3, which reports the time taken to train GNNs for node classification, with a varying number of GNN layers. The results show that
> - The proposed forward learning approaches in Sections 3.1 and 3.2 train GCN and SAGE
> models much faster than backpropagation (up to 11 times) by exploiting the characteristics of greedy forward learning.
> - While the training with forward learning approaches takes longer than BP in some other cases, the increase in training time in such cases is modest in comparison to BP (e.g., ~1.6-1.9 times on GitHub).
> - The single-forward (SF) approach improves the training efficiency of the forward-forward based approaches.
>
>
> > Q6. In Table 2, why does SF-Top-To-Loss achieve state-of-the-art performance with one layer, but the performance degrades with more layers? An explanation is needed.
>
> In those failure cases of SF-Top-To-Loss, adding additional layers seems to introduce a larger noise to the node embeddings learned by lower layers, leading to a performance decrease. Depending on the dataset and GNNs, this may be because the upper layers learn bad representations at the beginning of the training, and the lower layers, which incorporate the embeddings from the upper layers for optimization, fail to learn meaningful representations due to the influence of the noise from above, and get stuck in the local optima. In such a case, suboptimal representations from lower layers may in turn make the learning at the upper layers more difficult.
>
>
> > Q7. In Table 2, the best-performing method varies significantly across datasets. Providing an intuition for this variation would be beneficial.
>
> While there is no consistent winner in Table 2, the best results are achieved by the SF-Top-To-Input (or the SF) approach in many cases. CiteSeer is an exception, where FF-LA performs better than the single forward (SF)-based approaches. We think that in this case, the label information, which is directly appended to the input features, provides a stronger and more direct signal to infer the unknown labels of testing nodes than in other datasets, e.g., due to the tendency of nearby nodes sharing the same labels.
> At the same time, we want to bring two additional aspects into the evaluation of different learning approaches, namely, the memory usage and training efficiency. Considering those two factors along with prediction accuracy, the SF-Top-To-Input, or the SF outperforms the FF-based methods, and provides an overall best performance.
>
>
> > Q8. In Table 3, the memory of some proposed methods is higher than BP (even 20 times higher in Table 3 (c) GAT). Would this be a significant bottleneck?
>
> In the paper, we present variants of forward-only graph learning paradigms, making incremental improvements such as learning via a single forward pass, and top-down signal incorporation. In that sense, we recommend using single forward pass-based approaches in practice, as they achieve similar, and often, better accuracy while requiring less memory and time for learning than the approaches that directly extend the forward-forward (FF) algorithm (Sec 3.1). At the same time, the FF-based variant provides a building block, such as the virtual node idea, which we use for the follow-up single-forward variants.
> With respect to the FF-based approach, its high memory usage can indeed be a bottleneck, especially for memory-intensive GNNs, and graphs with a large number of classes, as its memory usage as well as the run time increase in proportion to the number of negative inputs that need to be generated to achieve a good performance. The improvement achieved by the single-forward approaches over the FF-based ones shows the benefits of our proposed method.

---

### Official Review · Reviewer_BuBQ · 2023-10-21

**Soundness:** 3 good
**Presentation:** 2 fair
**Contribution:** 2 fair
**Rating:** 6
**Confidence:** 4

**Summary:**

The de-facto standard algorithm for training Graph Neural Networks (GNNs) is *Backpropagation*. Despite several advantages, the need to backpropagate gradients through the neural architecture hinders its scalability whenever the architecture depth increases. The recent Forward-Forward (FF) approach by Hinton et al. has inspired several works that aim at local/forward-only learning procedures. In this work, the authors propose ForwardGNN, which investigates FF in the context of GNNs for node classification and link prediction tasks, as well as and proposing a novel approach that requires only a single forward pass to learn GNNs.

**Strengths:**

The paper is well written and structured. The proposed approach is original, given that it extends the **FF** approach to GNNs by proposing a forward-only mechanism that avoids multiple forward passes for the positive and negative samples,  that would be required by standard FF.
Moreover, the interesting incorporation of top-down signals from upper layers is a clever intuition.

**Weaknesses:**

There are some details of the approach that have not been clearly described.
In the case of the approaches that leverage virtual nodes (Sections 3.1-bottom and 3.2): the authors specify that the graph topology is enriched by such virtual nodes. It is not clear to me wheter such virtual nodes are processed as standard nodes by the GNNs -- e.g. they need initial nodal features $h_i^{(0)}$ and both receive and send message towards neighbors -- or they are solely used as *receiver* nodes, e.g. they only have incoming edges-- in order to compute their representation $c_k^{(\ell)}$.

The role of the virtual nodes in the graph topology raises another question: if I did get it correctly, there are as many virtual nodes as classes. Hence, many nodes (depending on the graph scale) will be connected to the same virtual node forming a bottleneck. This approach seem to be very prone to the issue of over-squashing [1,2]. What happens when the number of classes is very low with respect to the graph scale (with millions of nodes)? An analysis on this could improve the paper contribution.

The experimental setup analyzes some competitors that were not devised for GNN in the tasks of node classification and link prediction. Given that the **GFF** model by Paliotta et al. [3] is explicitly devised for GNNs, why did the authors not compare with **GFF**? Is the proposed approach compatible with the Graph classification task?

Regarding related work, there are some works that proposed alternative local rules for learning in GNNs that depart from the BackProp approach [4, 5]. Describing differences and advantages could help the reader in understanding the paper contributions.


*Minors*
The authors refer to Alg. 1, 2, 3 that are not in the main paper (it should be clarified).

[1] Uri Alon and Eran Yahav. On the bottleneck of graph neural networks and its practical implications.
arXiv preprint arXiv:2006.05205, 2020.

[2] Francesco Di Giovanni, Lorenzo Giusti, Federico Barbero, Giulia Luise, Pietro Lio, and Michael M
Bronstein. On over-squashing in message passing neural networks: The impact of width, depth,
and topology. In International Conference on Machine Learning, pp. 7865–7885. PMLR, 2023.

[3] Daniele Paliotta, Mathieu Alain, Balint Me, and Francois Fleuret. Graph neural networks go forward-forward. CoRR, abs/2302.05282, 2023

[4] Tiezzi, Matteo, et al. "Deep constraint-based propagation in graph neural networks." IEEE Transactions on Pattern Analysis and Machine Intelligence 44.2 (2021): 727-739.

[5] H. Dai, Z. Kozareva, B. Dai, A. J. Smola, and L. Song, “Learning steady-states of iterative algorithms over graphs,” in Proc. Int.
Conf. Mach. Learn., 2018, pp. 1114–1122

**Questions:**

Please also refer to the **Weaknesses** section.

1) It is not clear to me wheter virtual nodes are processed as standard nodes by the GNNs -- e.g. they need initial nodal features $h_i^{(0)}$ and both receive and send message towards neighbors -- or they are solely used as *receiver* nodes, e.g. they only have incoming edges-- in order to compute their representation $c_k^{(\ell)}$.

2) The virtual node approach seem to be very prone to the issue of over-squashing [1,2]. What happens when varying the number of classes with respect to the graph scale (with millions of nodes)? An analysis on this could improve the paper contribution.

3) The experimental setup analyzes some competitor algorithms that were not devised for GNNs, in the tasks of node classification and link prediction. Given that the **GFF** model by Paliotta et al. [3] is explicitly devised for GNNs, why did the authors not compare with **GFF**? Is the proposed approach compatible with the Graph classification task?

4) The authors analyzed the memory impact of the proposed method. What about the time complexity/execution timings?  And what are the model ability to scale to bigger graphs?

---

> ### Author Response · Authors · 2023-11-23
> **Response to Reviewer BuBQ**
>
> We appreciate your thoughtful comments. Please find our response to your comments below. We have updated the paper where necessary in light of your comment.
>
> > It is not clear to me wheter virtual nodes are processed as standard nodes by the GNNs -- e.g. they need initial nodal features and both receive and send message towards neighbors -- or they are solely used as receiver nodes, e.g. they only have incoming edges-- in order to compute their representation.
>
> The virtual nodes are indeed processed in the same way as the standard nodes by GNNs, and we randomly initialized the input node features. In the main text, the results of the proposed approaches that employ virtual nodes were obtained with the augmented graph where real nodes and virtual nodes are connected via bidirectional edges (i.e., in both ways between the real and virtual nodes). In Section 3.1, we highlighted the sentences that describe the use of these bidirectional edges.
>
>
> > The virtual node approach seem to be very prone to the issue of over-squashing [1,2]. What happens when varying the number of classes with respect to the graph scale (with millions of nodes)? An analysis on this could improve the paper contribution.
>
> Thank you for the insightful comment. Since the virtual nodes, which are connected via bidirectional edges, are processed in the same way as the real nodes, the information aggregation and propagation via those virtual nodes may potentially adversely impact the performance, as pointed out by the comment. So we evaluated a different approach to utilize virtual nodes in the same problem setup, which is to connect the real and virtual nodes via unidirectional edges (i.e., in one way from the real to the virtual nodes), such that virtual nodes only receive information from the real nodes, without sending the aggregated message back to them.
>
> In Figure 9 and Appendix E.5 (highlighted in brick red), we report the node classification performance when the edges between the real and virtual nodes are bidirectional and unidirectional. Results show that using unidirectional edges performs better than using bidirectional edges in many cases, while both perform similarly in the other cases. In the final version of this paper, we will include more comprehensive results using the unidirectional augmented edges.
>
>
> > The experimental setup analyzes some competitor algorithms that were not devised for GNNs, in the tasks of node classification and link prediction. Given that the GFF model by Paliotta et al. [3] is explicitly devised for GNNs, why did the authors not compare with GFF? Is the proposed approach compatible with the Graph classification task?
>
> Although the GFF model was designed for GNNs, it could not be included in the evaluation since GFF is specifically designed for the graph classification task, and cannot be used in its original form for the node classification and link prediction tasks, which is the focus of our work.
> However, at the same time, our proposed adaptation of the original FF algorithm (Hinton, 2022) described in Section 3.1 can be considered as a way to modify the GFF model to be applicable for tasks such as node classification, based on the forward-forward learning mechanism that GFF employs. Also, in the paper, we make further improvements to this type of adaptation of the FF algorithm by enabling the forward graph learning via a single forward pass (Sec 3.2), and the incorporation of the top-down signals (Sec 3.3), both of which are not possible with a straightforward adaptation of the GFF model. Experimental results show that the proposed improvements in Secs. 3.2 and 3.3 lead to similar, and often better prediction performance, while taking a smaller amount of memory and computation time than the approach directly based on FF (e.g., the aforementioned adaptation of GFF, or Sec. 3.1).
>
> The proposed approach can indeed be made to address the graph classification task via a single-forward pass, as opposed to the multiple forward passes required by GFF. To that end, we will need to replace the virtual node idea, which is used to obtain representative nodes in our current work, with a new technique that can learn representatives at the graph level. Including the development of an effective method to learn graph representatives in the forward learning setup, investigating the potential of forward learning for different types of GL tasks (e.g., graph classification) is one of the future plans for this work.

---

> ### Author Response · Authors · 2023-11-23
> **Response to Reviewer BuBQ (Cont.)**
>
> > The authors analyzed the memory impact of the proposed method. What about the time complexity/execution timings? And what are the model ability to scale to bigger graphs?
>
> To evaluate the training efficiency of different learning approaches, we added additional results in Table 6 in Appendix E.3 (marked in blue), which reports the time taken to train GNNs for node classification, with a varying number of GNN layers. The results show that
> - The proposed forward learning approaches in Sections 3.1 and 3.2 train GCN and SAGE
> models much faster than backpropagation (up to 11 times) by exploiting the characteristics of greedy forward learning.
> - While the training with forward learning approaches takes longer than BP in some other cases, the increase in training time in such cases is modest in comparison to BP (e.g., ∼1.6-1.9 times on GitHub).
> - The single-forward (SF) approach improves the training efficiency of the forward-forward based approaches.
> These results show that the forward learning-based GNN training can scale up to large graphs, enabling faster and more memory efficient GNN training on large-scale graphs.
>
>
> > Regarding related work, there are some works that proposed alternative local rules for learning in GNNs that depart from the BackProp approach [4, 5]. Describing differences and advantages could help the reader in understanding the paper contributions.
>
> Thank you for the references. In Appendix D, we added a discussion on alternative learning approaches for training GNNs. Specifically, for the mentioned approaches [4, 5], we added the following discussion (marked in brick red).
>
> Carreira-Perpinan & Wang (2014) and Taylor et al. (2016) proposed alternative approaches to backpropagation (BP) for learning neural networks (NNs), where the learning of NNs is cast as a constrained optimization problem in the Lagrangian framework, with neural computations expressed as constraints. These approaches are inherently local, in the sense that the gradient computation relies on neighboring neurons, not on the BP over the entire network. However, they are not scalable due to the high memory requirement, and thus are inapplicable to large-scale problems.
>
> Based on this Lagrangian-based approach, Tiezzi et al. (2020; 2022) presented an alternative
> method for learning GNNs. Specifically, they simplified the learning process of the GNN* model
> (Scarselli et al., 2009), a representative recurrent GNN model, which performs graph diffusion repeatedly until it converges to a stable fixed point. The LP-GNN proposed in Tiezzi et al. (2020; 2022) adopts a mixed strategy to enable a more efficient training of GNN*, which employs a constraint-based propagation scheme, thereby avoiding the explicit computation of the fixed point of the diffusion process in GNN*, while using BP in updating the NNs that model the state transition and output functions. Accordingly, although LP-GNN departs from the usual BP-based GNN training, it still relies on BP for learning GNNs. By contrast, ForwardGNN is free from BP as a forward-only learning scheme for GNNs.
>
> Stochastic steady-state embedding (SSE) (Dai et al., 2018) presents another alternative learning algorithm, in the context of learning steady-state solutions of iterative algorithms over graphs, e.g., PageRank scores and connected components. To obtain steady-state solutions efficiently, SSE proposes a stochastic fixed-point gradient descent process, inspired by the policy iteration in reinforcement learning. While SSE presents an alternative learning approach, SSE is not, per se, a general training scheme for GNNs like ForwardGNN, as its goal is to learn an algorithm achieving steady-state solutions of iterative graph algorithms, which greatly differs from the task of GNN training ForwardGNN addresses.
>
>
> > The authors refer to Alg. 1, 2, 3 that are not in the main paper (it should be clarified).
>
> Thank you for pointing this out. We clarified in Section 3 (marked in brick red) that those algorithm listings are provided in the Appendix.

---

### Official Review · Reviewer_ztUM · 2023-11-02

**Soundness:** 3 good
**Presentation:** 3 good
**Contribution:** 3 good
**Rating:** 6
**Confidence:** 2

**Summary:**

Traditional training of GNNs relies on the backpropagation (BP) algorithm, which imposes certain constraints that limit scalability, parallelism, and flexibility in learning. To overcome these limitations, the authors propose FORWARDGNN, inspired by the forward-forward (FF) algorithm used in image classification. FORWARDGNN extends FF to work with graph data, eliminating the need for generating negative inputs and allowing layer-wise local forward training. The new method enables each layer to learn from both bottom-up and top-down signals without relying on error backpropagation. The paper demonstrates the effectiveness and generality of FORWARDGNN through experiments on five real-world datasets and three GNNs, showing that it performs on par or better than BP on link prediction and node classification tasks while being more memory efficient.

**Strengths:**

Forward-Forward learning proposed by Hinton is a very new and interesting research topic. This paper adopts that in GNN setup, and propose an alternative ( Single forward ) which only runs a single forward pass.


Experiments are very comprehensive, including both effectiveness and training efficiency.

**Weaknesses:**

Overall I think this is an interesting paper. Given the Forward-Forward Learning is a very new concept, this work is an interesting trial on the direction.

One question is can the proposed forward-only method only work for graph learning? Or it can be generalized to other tasks? If yes, better to show such results to make this work more solid; if not, better to explain clearly the assumption and some unique properties of graph to make this method work.

**Questions:**

see weakness.

How is this paper different from: "Decoupled Greedy Learning of Graph Neural Networks"

---

> ### Author Response · Authors · 2023-11-23
> **Response to Reviewer ztUM**
>
> We appreciate your thoughtful comments. Please find our response to your comments below. We have updated the paper where necessary in light of your comment.
>
>
> > One question is can the proposed forward-only method only work for graph learning? Or it can be generalized to other tasks? If yes, better to show such results to make this work more solid; if not, better to explain clearly the assumption and some unique properties of graph to make this method work.
>
> In the proposed forward-only method, the top-down signal incorporation (Section 3.3) does not rely on utilizing the characteristics of graphs or GNNs, so it can be applied to the forward learning of different types of tasks. Also, the idea of replacing the two forward passes of the forward-forward (FF) algorithm with a single forward pass (Section 3.2) is not particularly tied to GNNs, so it could be applicable in other contexts depending on the tasks.
>
> By contrast, some other parts of the proposed approach (such as the virtual node-based graph structure extension technique used in Sections 3.1 and 3.2) make a specific use of graph data, and thus would be hard to use for non-graph tasks in the original form. Still, those techniques are applicable to any types of graphs, as they do not make particular assumptions on the types or characteristics of graphs.
>
> In sum, the answer to this question is mixed with yes and no, and accordingly, applying the entirety of the proposed forward-learning method to different tasks might be hard. Extending the generalizable parts of the proposed method to other tasks is one of the future plans for this work.
>
>
> > How is this paper different from: "Decoupled Greedy Learning of Graph Neural Networks"
>
> In short, the layer-wise learning presented in "Decoupled Greedy Learning of Graph Neural Networks" (Wang et al., arXiv 2020) does not achieve the forward-only learning of GNNs, which is the goal of the proposed ForwardGNN, as well as the forward-forward algorithm (Hinton, 2022), a forward-only learning method recently proposed as a biologically plausible alternative to the backpropagation (BP). Also, the aforementioned layer-wise learning fails to learn from the top-down signal (Section 3.3) that ForwardGNN incorporates into the learning process. In Appendix D, we added the following, more detailed discussion of layer-wise learning approaches for GNNs (marked in blue).
>
> "A layer-wise learning idea closely related to our work first appeared in the vision domain (Belilovsky et al., 2019), which proposed a greedy layer-wise learning technique for deep convolutional neural networks (CNNs), and obtained promising results on large-scale datasets such as ImageNet. In Belilovsky et al. (2019), a CNN model gets trained progressively and greedily, i.e., layer by layer from the bottom to the top. To enable a layer-wise training, they formulate and solve an auxiliary problem for each layer, where an auxiliary classifier is attached to the layer, forming a block, and the original CNN layer and the auxiliary classifier in the block are jointly optimized in terms of the training accuracy of the auxiliary classifier. You et al. (2020) extended this approach for a layer-wise training of GNNs, in which the CNN layers are replaced with the GNN layers. Furthermore, You et al. (2020) streamlined the graph convolution operation by decoupling the feature aggregation and transformation, and introduced a learnable RNN controller, which automatically decides when to stop the training of each layer via reinforcement learning. A follow-up work presented by Wang et al. (2020) further improved the efficiency of layer-wise GNN training of You et al. (2020) via the proposed parallel training and lazy update scheme. While these methods train GNNs layer by layer, they still rely on BP, since each block is a multi-layer architecture, and as a result, the error derivatives are propagated across the layers in the block for optimization. By contrast, ForwardGNN operates in the forward-only learning regime, which is the focus of this work. Further, their optimization framework allows only bottom-up signal propagation, and does not provide a way to incorporate the top-down signals. Thus, using the above layer-wise methods, the training of each layer cannot be informed by what the upper layers have learned as in ForwardGNN. Also, these studies only investigate the node classification task, and miss out the other fundamental graph learning task of link prediction. In this work, we systematically explore the potential of forward-only learning for both node classification and link prediction tasks."

---

### Official Review · Reviewer_yhLp · 2023-11-05

**Soundness:** 3 good
**Presentation:** 3 good
**Contribution:** 3 good
**Rating:** 8
**Confidence:** 4

**Summary:**

This paper is an application of forward forward learning on graph neural networks. Authors modify the forward forward learning algorithm to train GNNs with several novel designs including node labels, virtual nodes, single forward pass, and the top-down signals. In summary, this paper is a good practice by applying forward forward algorithm to GNN learning and provides sufficient technical contribution with sound experiments. I raise a marginally accept for the lack of important experiments.

**Strengths:**

1. Authors systematically investigate the forward learning algorithm on GNNs.
2. This paper provides several technical contributions to forward forward algorithm, which is inspiring and important.
3. Extensive experiments have been carried out to prove their effectiveness.

**Weaknesses:**

1. Motivation is not well illustrated
2. Time efficiency is not analyzed

**Questions:**

1. From my point of view, the motivation of this paper is to apply the forward forward algorithm to GNN training. They do not mention what the problem is with the current BP training method on GNN. The three points (scalability, parallelism, and flexibility) they mentioned do not seem to be graph-related. I suggest finding a stronger motivation in the introduction, such as what the problem FF wants to solve on graph learning.
2. By training layer by layer, FF can be seen as a time-for-space algorithm. In the experiment, authors only show the limited space utilization without mentioning the overhead training time. I think it is inappropriate.
3. In the node classification task, authors randomly selected 64% for training, which is not the common practice in graph learning. In most cases, we only select a small percentage for training, as in the GCN, GAT, and SAGE paper author mentioned. I think more experiments on limited training data are needed as their proposed algorithm seems sensitive to that.

---

> ### Author Response · Authors · 2023-11-23
> **Response to Reviewer yhLp**
>
> We appreciate your thoughtful comments. Please find our response to your comments below. We have updated the paper where necessary in light of your comment.
>
>
> > From my point of view, the motivation of this paper is to apply the forward forward algorithm to GNN training. They do not mention what the problem is with the current BP training method on GNN. The three points (scalability, parallelism, and flexibility) they mentioned do not seem to be graph-related. I suggest finding a stronger motivation in the introduction, such as what the problem FF wants to solve on graph learning.
>
> Thank you for the comment. The BP-based GNN training is limited by the same constraints of BP for other types of NNs as discussed in the first part of the Introduction section. While the forward-forward (FF) algorithm provides a promising alternative to BP, existing FF approaches fail to be an effective alternative to BP in learning GNNs, which motivates our goal of developing an effective and efficient forward-only learning approach for GNNs.
>
> More specifically, to make the problems and the goal of this work more clear, we revised the Introduction section as follows (marked in orange):
> * “The training of GNNs is limited by the same constraints of BP as discussed above,
> which restrict the scalability and flexibility of model training, as well as rendering the learning process biologically implausible. While FF and its recent variants (Hinton, 2022; Zhao et al., 2023; Lee & Song, 2023; Paliotta et al., 2023) saw promising results, they fail to provide an effective and efficient alternative for learning GNNs for the following reasons. First, the FF algorithm involves generating negative data, which is a task-specific process (e.g., handcrafting masks for images with varying long range correlations, and manipulating images via overlaying class labels), and needs to be defined to be used for new tasks. Existing FF methods are mainly designed for image classification, and thus are difficult or ineffective for handling graph data. Second, the current way FF generates negative data is not scalable as it takes an increasing amount of memory and computation time as more negative samples are used.  To date, the potential of FF for the fundamental graph learning (GL) tasks, i.e., node classification and link prediction, has not been studied, and it remains to be explored how effectively FF or alternative forward learning procedures can train GNNs for fundamental GL tasks.”
>
> > By training layer by layer, FF can be seen as a time-for-space algorithm. In the experiment, authors only show the limited space utilization without mentioning the overhead training time. I think it is inappropriate.
>
> To evaluate the training efficiency of different learning approaches, we added additional results in Table 6 in Appendix E.3 (marked in blue), which reports the time taken to train GNNs for node classification, with a varying number of GNN layers. The results show that
> - The proposed forward learning approaches in Sections 3.1 and 3.2 train GCN and SAGE
> models much faster than backpropagation (up to 11 times)  by exploiting the characteristics of greedy forward learning.
> - While the training with forward learning approaches takes longer than BP in some other cases, the increase in training time in such cases is modest in comparison to BP (e.g., ~1.6-1.9 times on GitHub).
> - The single-forward (SF) approach improves the training efficiency of the forward-forward based approaches.

---

> ### Author Response · Authors · 2023-11-23
> **Response to Reviewer yhLp (Cont.)**
>
> > In the node classification task, authors randomly selected 64% for training, which is not the common practice in graph learning. In most cases, we only select a small percentage for training, as in the GCN, GAT, and SAGE paper author mentioned. I think more experiments on limited training data are needed as their proposed algorithm seems sensitive to that.
>
> We added Appendix E.4 (marked in blue) to evaluate how well the proposed forward learning approach performs in comparison to BP when given a small amount of training data. Specifically, we use three citation networks with limited training data, which have been widely used in earlier works, such as the GCN paper (“Semi-Supervised Classification with Graph Convolutional Networks”). Table 7 shows the number of training/validation/testing nodes for each dataset used for evaluation. In Figure 8, we show the classification accuracy versus memory usage increase, obtained with the single forward (SF) and backpropagation (BP) algorithms on three graphs with limited training data.
> - The proposed SF (empty symbols) perform similarly to BP (filled symbols), or outperforms BP in a few cases, when four GNN layers are used, which is similar to the results obtained in the setup with richer training data (Figure 2a).
> - When two layers are used, BP mostly outperform SF, which is different from the results in the counterpart in Figure 2a(1), where BP and SF performed comparably to each other. The contrastive single forward learning mechanism tends to require more training data than BP to arrive at a similar level of predictive accuracy, but the gap closes as more layers are used.
> - These results point to an interesting research direction to further improve the performance of forward-only GNN training in the regime of sparse training data.
>
> In the final version of this paper, we will include more comprehensive results on datasets with limited training data.

---

> > ### Comment · Reviewer_yhLp · 2023-12-04
> > **Increase score to 8**
> >
> > Great change. I have raised my score to 8

---

### Meta-Review · Area_Chair_kHUT · 2023-12-08

**Metareview:**

This paper proposes a forward-only learning algorithm for graph neural networks. While it does not present significantly novel technical ideas, it is a solid contribution, as it also shows the efficiency of learning by not performing BP. Reviewers are generally positive overall.

**Justification For Why Not Higher Score:**

Technical contributions are not too significant, given the motivation to apply FF to GNN.

**Justification For Why Not Lower Score:**

The idea combining GNN + FF will attract interests in the community.

---

### Decision · Program_Chairs · 2024-01-16

Accept (poster)